# Development of efficient aqueous organic redox flow batteries using ion-sieving sulfonated polymer membranes

Chunchun Ye[1,2,6], Anqi Wang [1,6], Charlotte Breakwell[3], Rui Tan[1], C. Grazia Bezzu [2], Elwin Hunter-Sellars[1], Daryl R. Williams[1], Nigel P. Brandon[4], Peter A. A. Klusener [5], Anthony R. Kucernak [3], Kim E. Jelfs [3], Neil B. McKeown [2✉] & Qilei Song [1✉]

Redox flow batteries using aqueous organic-based electrolytes are promising candidates for developing cost-effective grid-scale energy storage devices. However, a significant drawback of these batteries is the cross-mixing of active species through the membrane, which causes battery performance degradation. To overcome this issue, here we report size-selective ion-exchange membranes prepared by sulfonation of a spirobifluorene-based microporous polymer and demonstrate their efficient ion sieving functions in flow batteries. The spirobifluorene unit allows control over the degree of sulfonation to optimize the transport of cations, whilst the microporous structure inhibits the crossover of organic molecules via molecular sieving. Furthermore, the enhanced membrane selectivity mitigates the crossover-induced capacity decay whilst maintaining good ionic conductivity for aqueous electrolyte solution at pH 9, where the redox-active organic molecules show long-term stability. We also prove the boosting effect of the membranes on the energy efficiency and peak power density of the aqueous redox flow battery, which shows stable operation for about 120 h (i.e., 2100 charge-discharge cycles at 100 mA cm$^{-2}$) in a laboratory-scale cell.

[1] Department of Chemical Engineering, Imperial College London, London SW7 2AZ, UK. [2] EaStCHEM, School of Chemistry, University of Edinburgh, Edinburgh EH9 3FJ, UK. [3] Department of Chemistry, Molecular Sciences Research Hub, Imperial College London, London W12 0BZ, UK. [4] Department of Earth Science and Engineering, Imperial College London, London, UK. [5] Shell Global Solutions International B.V., Shell Technology Centre Amsterdam, Grasweg 31, 1031 HW Amsterdam, The Netherlands. [6] These authors contributed equally: Chunchun Ye, Anqi Wang. ✉email: neil.mckeown@ed.ac.uk; q.song@imperial.ac.uk

Large-scale long-duration energy storage technologies are required to facilitate the transition away from electricity generation using fossil fuels to renewable but intermittent energy sources such as solar and wind power. Redox flow batteries (RFBs) show great promise for grid-scale energy storage owing to the long discharge duration at rated power, scalable energy and power density, high power output, and the potential for long operational lifetime[1–4]. In order to develop RFBs for gigawatt scale energy stage[5], low-cost redox species have received significant attention in recent years, such as transition-metal complexes, organic molecules[6–9], and polymers[10]. Unfortunately, the lifetime of these chemicals in RFBs is limited by degradation of the redox-active species under the challenging electrochemical conditions, particularly highly alkaline electrolytes (pH > 14)[11]. Recent studies have developed functionalized organic molecules with high solubility in less chemically harsh pH conditions (e.g., >0.5 M solubility at pH 12)[12], resulting in lower capacity degradation[12–14]. Despite these advances, these organic RFBs use two different electrolytes and then the crossover of redox species through the membrane separator becomes the key factor in battery capacity decay. The optimization of RFB performance require fit-for-purpose membranes to match with redox species and electrolyte solutions. Therefore, the development of membranes meeting all the key criteria for flow batteries, including high ionic conductivity, high molecular selectivity, high stability as well as ease of manufacture at low cost, has become an important challenge in the flow battery field[15].

Membranes comprising subnanometer channels (<1.0 nm) are widely used in gas separation, water purification, and devices for energy generation and storage[16,17]. The design and control over channel architecture and functionality for selective mass transport are key to improving membrane properties of permeability and selectivity for optimal device performance[18]. In electrochemical devices, such as fuel cells, flow batteries and electrolyzers, ion-exchange membranes are crucial components that determine the energy efficiency, power output, and lifetime of these devices[19,20]. Commercial perfluorinated ion-exchange membranes (e.g., Nafion) are currently widely used, however, their high cost and limited selectivity impede to a significant extent the large-scale application of RFB technology. A wide range of hydrocarbon-based polymer ion-exchange membranes have been developed in the past three decades[19,21] (Supplementary Fig. 1), however, the microphase separation between flexible hydrophobic backbones and hydrophilic side chains is very challenging to control, resulting in ill-defined ion transport channels[19].

More recently, polymers of intrinsic microporosity (PIMs) have shown great promise for developing new-generation gas separation and ion transport membranes with combined functionalities of molecular sieving and high permeability. PIMs consist of rigid and contorted macromolecules which lead to inefficient packing in the solid state hence generating high free volume that provides interconnected micropores less than 2 nm, which are considered as microporous materials according to definition of International Union of Pure and Applied Chemistry (IUPAC)[22–26]. Hydrophobic PIM polymers have been demonstrated as size-selective separators in Li-S batteries[27], non-aqueous organic redox flow batteries[28], and all-vanadium flow batteries[29]. Recent reports show that incorporating ionizable functional groups such as amidoxime, which can be deprotonated in alkaline electrolyte solutions[30,31], produces cation exchange PIM membranes that give moderate to high ion conductivity (approaching $10^{-2}$ S cm$^{-1}$ at 80 °C) in pH 14 alkaline flow batteries. Unfortunately, these membranes show poor ionic conductivity at lower pH (e.g., $10^{-4}$ S cm$^{-1}$ even at 80 °C in pH 7 NaCl solution)[31]. Hence, it is highly desirable to develop ion-conductive PIMs that possess negatively charged functionality at neutral pH such as sulfonate groups. However, a previous report describes the challenges in attempting to sulfonate PIM-1[32], the most commonly investigated PIM prepared from commercially available spirocyclic biscatechol and tetrahalo monomers via dibenzodioxin-forming polymerization. All common sulfonating agents result in hydrolysis of the nitrile groups and degradation of the polymer so that it was impractical to form membranes. This outcome is unsurprising due to the hindered aromatic sites within PIM-1 and the bulky sulfonic acid group. Recently, we also reported sulfonated polymer membranes for aqueous RFBs and fuel cells[33]. However, these membranes still suffer from the trade-off between ionic conductivity and selectivity. A major scientific challenge remains to be addressed, i.e., achieving a membrane with high ionic conductivity (>$10^{-2}$ S cm$^{-1}$) at slightly-alkaline pH (e.g., pH 9) and ambient operation temperature (<30 °C) for the long-term stability of organic redox species while maintaining high ionic and molecular selectivity.

In this work, we report ion-exchange membranes with sub-nanometer ion transport pathways derived from PIMs and demonstrate their improved performance in aqueous organic RFBs operated at pH 9 (Fig. 1a). To demonstrate the concept, we used a spirobifluorene-based PIM polymer (PIM-SBF) as a model precursor and performed precisely controlled sulfonation modification using the mild silyl-protected sulfonation agent to introduce cation-conducting functionality within the well-defined micropores while minimizing side reactions (Supplementary Fig. 2). The resulting sulfonated PIMs (sPIM-SBFs) retain high structural rigidity and interconnected subnanometer pores with pore dimensions targeted for blocking redox-active small molecules and facilitating salt ion transport (Fig. 1b). Sulfonated PIM membranes enable rapid transport of small salt cations and inhibits crossover of the large anions and anionic redox-active molecules, resulting in significantly improved performance in comparison to commercial ion exchange membranes and ion sieving membranes from state-of-the-art porous materials. Importantly, the membranes demonstrate high conductivity for aqueous electrolytes at pH 9, in which both redox-active anthraquinone and ferrocyanide show long-term stability. By pairing the PIM membranes with slightly-alkaline electrolytes, our membranes enable efficient and highly stable battery operations for about 120 h in laboratory scale flow cells, demonstrating a significantly improved lifetime of organic flow batteries.

## Results

**Synthesis and physicochemical characterizations of polymers and membranes**. The key structural design concept is achieved by incorporating negatively charged sulfonate groups into PIM polymers with a rigid and contorted backbone (Fig. 2a). We designed and synthesized sulfonated PIMs based on 9,9′-spir-obifluorene (SBF) aromatic building blocks that possess activated and sterically uncrowded sites for electrophilic sulfonation. Sulfonation was achieved on both SBF model compounds and PIM-SBF polymers[34,35] through a one-step functionalization reaction using trimethylsilyl chlorosulfonate (TMSCS) (Fig. 2b). On addition of TMSCS to a PIM-SBF polymer solution in chloroform, sulfonation occurred immediately with the sulfonated polymer precipitating from solution. The degree of sulfonation was precisely controlled by regulating the molar ratio of TMSCS and PIM-SBF repeating units in the reaction. Unlike other strong sulfonation agents, such as sulfuric acid or chlorosulfonic acid, TMSCS does not induce side reactions such as hydrolysis of the nitrile groups, backbone cleavage, and avoids crosslinking[36].

Characterization of the sulfonated polymers was performed using $^1$H, $^{13}$C nuclear magnetic resonance (NMR) and fourier transform infrared (FTIR) spectroscopies, gel

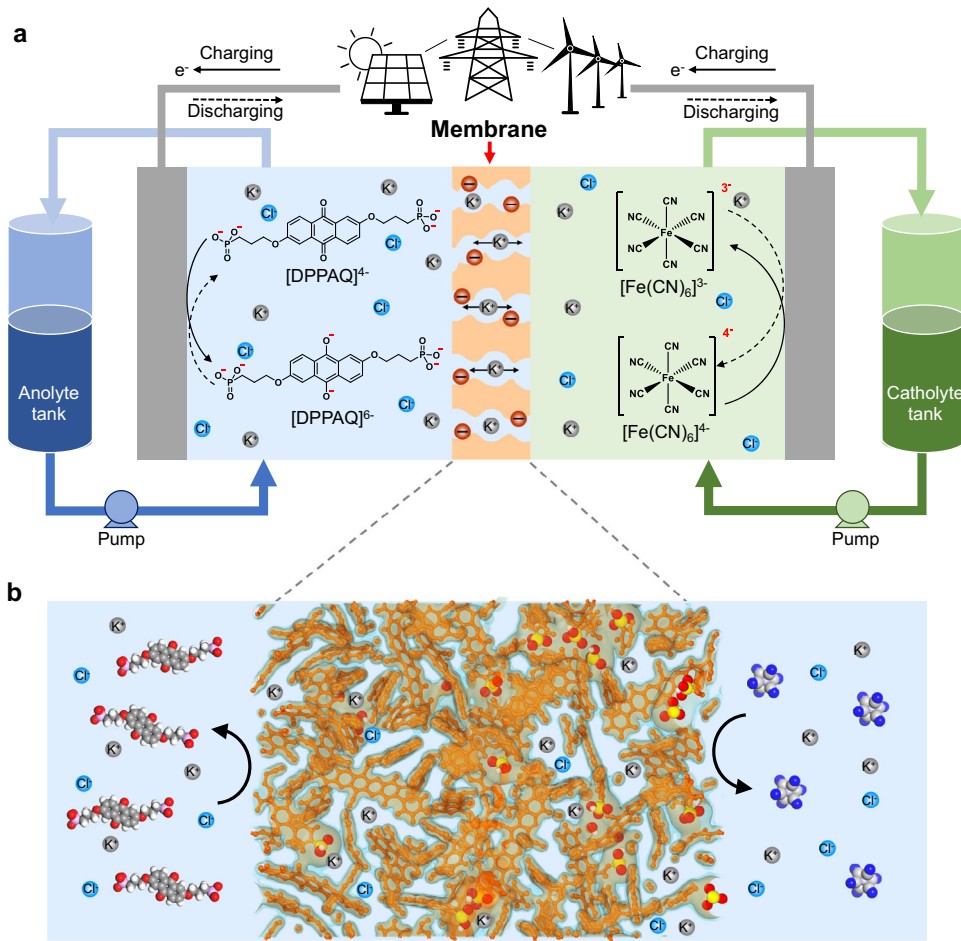

**Fig. 1 Ion-selective membranes for aqueous organic redox flow batteries. a** Schematics of an aqueous organic redox flow battery for grid-scale energy storage. Gray, blue and red spheres refer to $K^+$, $Cl^-$, and $SO_3^-$ groups, respectively. **b** Schematic showing the polymer ion-sieving membrane with subnanometer-sized pores that enable fast transport of charge-carrying ions while limiting the crossover of redox active species. Sulfur and oxygen atoms are colored as yellow and red, respectively.

permeation chromatography (GPC) and thermogravimetric analysis (TGA) analysis (Supplementary Figs. 3–7 and Supplementary Tables 1 and 2), helium pycnometry and titration measurements, as well as molecular simulations. The resulting polymers possess a range of sulfonate content, associated with ion exchange capacity (IEC) from 0.53–1.86 mmol $g^{-1}$ (termed as sPIM-SBF-0.53 to sPIM-SBF-1.86) and increased skeletal densities from 1.35 to 1.48 g $cm^{-3}$, which correlate smoothly to the molar equivalent of TMSCS used in the reaction mixture (Fig. 2c and Supplementary Table 3).

To understand the evolution of microporosity and formation of water channels, gas and water adsorption measurements were performed. $N_2$ adsorption isotherms at 77 K show that all sPIM-SBFs maintain microporosity and a high capacity for gas adsorption, although the apparent Brunauer–Emmett–Teller (BET) surface area decreases slightly from 692 $m^2$ $g^{-1}$ to 482 $m^2$ $g^{-1}$ on increasing the degree of sulfonation (Fig. 2d), while the bulky sulfonate groups would normally lead to remarkable loss of free volume for other porous materials such as covalent organic frameworks (COFs) and metal organic frameworks (MOFs)[37,38]. $CO_2$ adsorption at 273 K confirms their high adsorption capacity owing to the presence of micropores, with the derived pore size distributions indicating the predominance of subnanometer-sized micropores (Fig. 2e and Supplementary Fig. 8). Water uptake measured using dynamic vapor sorption (DVS) shows that adsorption correlates to the degree of sulfonation with up to 58 wt.% uptake for sPIM-SBF-1.86, which suggests the formation of a continuous network of water clusters within these sulfonate-functionalized micropores (Fig. 2f and Supplementary Fig. 9).

Molecular simulations of sPIM-SBFs in their dry state were performed to generate realistic structural models and analyse their porosity properties. Five independent models with different structural arrangements were established for each polymer system to ensure corresponding model was representative of its bulk material (Supplementary Fig. 10). As seen in cross sections of amorphous polymer cells (Fig. 2g), all sPIM-SBFs and PIM-SBF provide interconnected elements of free volume owing to the inefficient packing of contorted polymer chains. sPIM-SBFs exhibit a decrease in size of free volume elements as well as increase in skeletal densities on increasing the degree of sulfonation, which results from the occupation of free volume by bulky sulfonate groups (Supplementary Figs. 11–14 and Supplementary Tables 4–6). Simulated pore size distribution suggests that sPIM-SBFs possess narrow distributions of subnanometer-sized micropores with pore size centered around 0.5 nm, whereas PIM-SBF exhibits a broader pore size distribution centered around 0.7 nm (Supplementary Figs. 15 and 16). The evolution towards smaller and narrower free volume elements in sPIM-SBFs is consistent with observed experimental gas physisorption isotherms, which could offer confined ion transport at subnanometer scale.

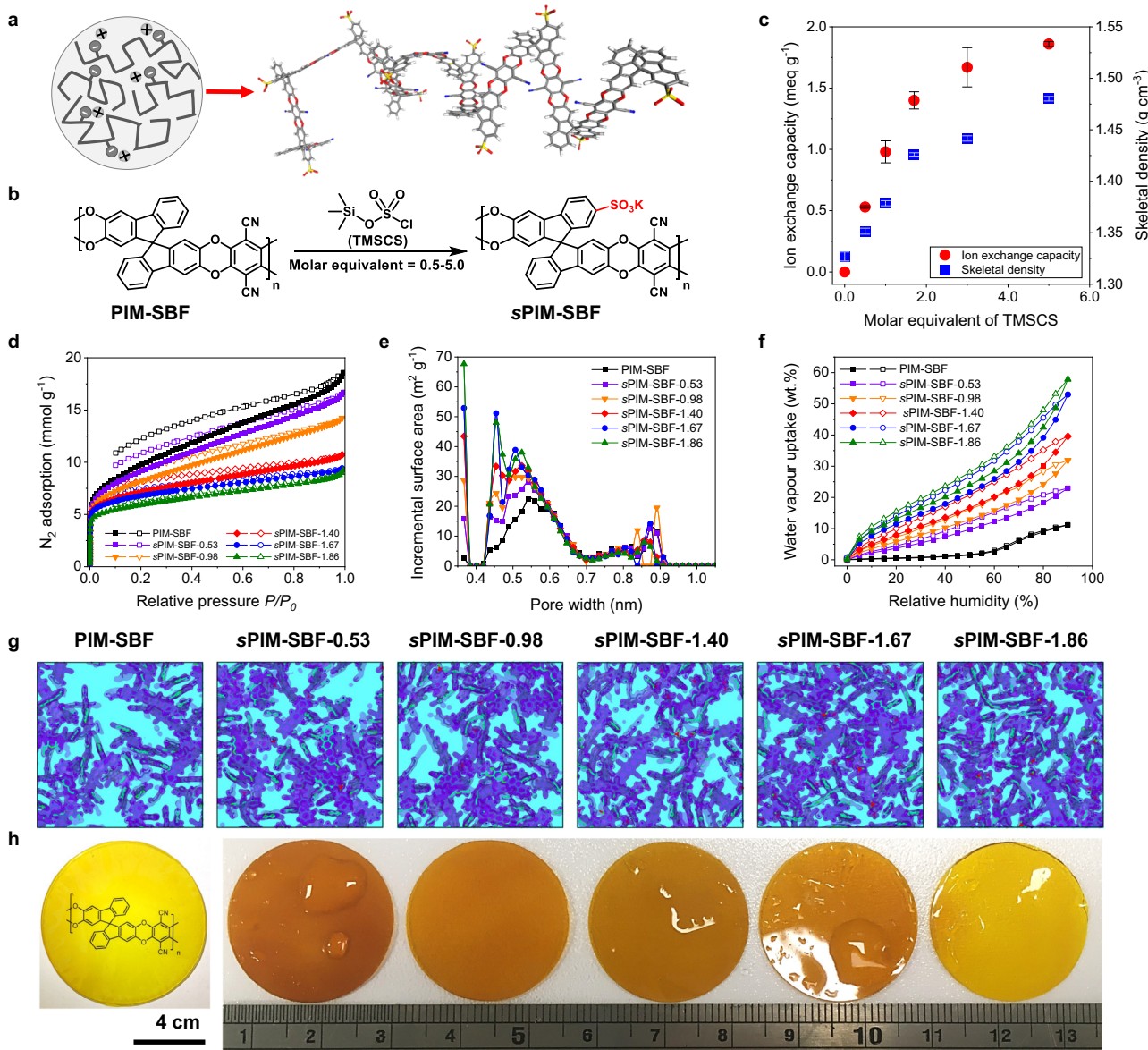

**Fig. 2 Synthesis and characterization of membranes. a** Schematic diagram and molecular model of a rigid and contorted PIM polymer chain with negatively charged sulfonate groups. **b** Synthetic route of sPIM-SBF polymers. **c** IEC and skeletal density of PIMs as a function of molar equivalent of TMSCS over PIM-SBF. Error bars are standard deviations derived from three IEC measurements based on three different samples, or a series of ten skeletal density measurements based on the same samples. **d** $N_2$ adsorption isotherms at 77 K. **e** Pore size distribution derived from $CO_2$ sorption based on DFT calculations. **f** Water vapor uptake isotherms at 298 K. **g** 1 nm-thick cross sections of the simulated polymer models in dry state. Purple shading highlights the isosurface of the polymer chain and light blue corresponding to the free volume. Sulfur atoms are colored in red and the polymer backbone in green. **h** Photograph of the PIM-SBF and sPIM-SBF membranes that corresponding to the modeling cell above each sample.

Combining the results from the experimental measurements and molecular simulation, we attribute the decrease of BET surface area and narrower pore size distribution to the bulkiness of the sulfonate groups and stronger inter-/intra-chain interactions. These sulfonated PIMs are readily soluble in polar organic solvents such as dimethylformamide, N-methyl-2-pyrrolidone and dimethyl sulfoxide and can be easily fabricated into mechanically robust self-standing films by casting from the solvent (Fig. 2h), with >30 MPa stress and >40% strain at break under ~50% relative humidity at room temperature (Supplementary Fig. 17 and Supplementary Table 7). Importantly for applications as membranes in aqueous systems, these films show a low linear swelling ratio, indicating the resistance to deterioration of mechanical robustness and expansion of pore dimension upon hydration (Supplementary Fig. 18). Despite their lower BET

surface areas and pore volumes, the sPIM-SBF membranes tend to swell following absorption of electrolyte solution. A greater degree of sulfonation enhances the degree of swelling of membranes and the expansion of the pore dimensions. Hence, moderately sulfonated sPIM-SBF is optimal to maintain the subnanometer size of electrolyte-filled micropores to provide the exquisite size-selectivity that favors charge carrier ions over larger redox-active species.

**Ionic conductivity**. To understand the relationship between microporosity and ion transport, ionic conductivity of membranes was measured by electrochemical impedance spectroscopy (EIS, Supplementary Fig. 19). As shown in Fig. 3a, the negatively charged subnanometer pores in the polymer membranes are filled with supporting electrolyte solution, forming interconnected,

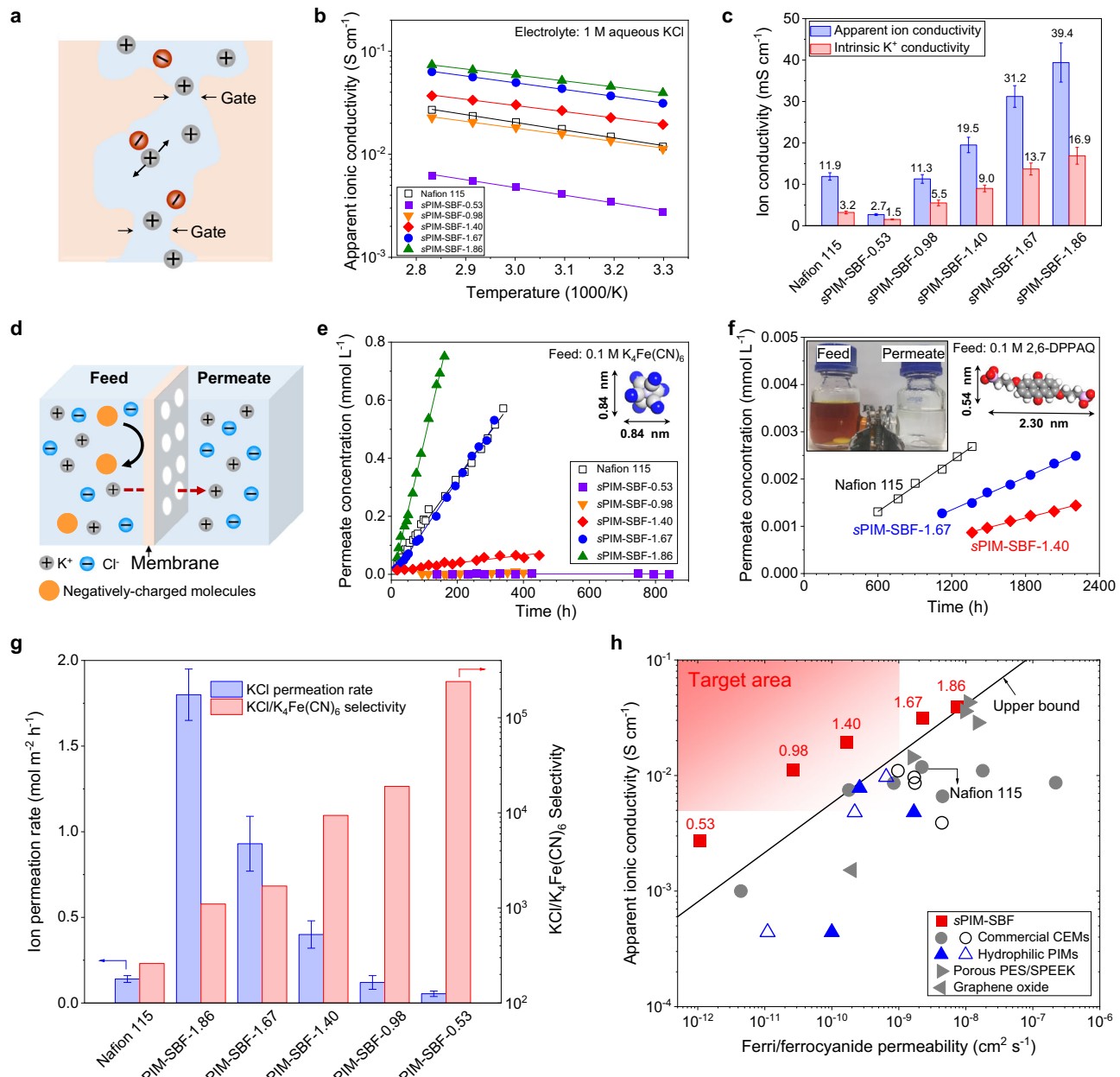

**Fig. 3 Ion conduction and molecular sieving properties. a** Schematic of the subnanometer pores (<1 nm) within sPIM-SBF membranes for fast ion transport. **b** Temperature dependence of apparent ionic conductivity measured by EIS in 1 M aqueous KCl. **c** Apparent ionic conductivity measured in 1 M aqueous KCl and intrinsic $K^+$ ionic conductivity measured in deionised water at 30 °C. Error bars are standard deviations derived from three ionic conductivity measurements based on three different membrane samples. **d** Schematic illustrating the transport of small $K^+$, $Cl^-$ ions through the ion-selective membrane and sieving of large redox-active molecules. **e** $K_4Fe(CN)_6$ and **f** 2,6-DPPAQ concentration in the permeate side of dialysis diffusion H-cells assembled with sPIM-SBF or Nafion 115 membranes as a function of time. Inset photograph showing the colors of feed and permeate solution in a H-cell assembled with sPIM-SBF-1.40 membrane after 3-month dialysis diffusion test. Data in **e** and **f** are linearly fitted to derive permeation rates and fitted lines are presented in the figures. **g** KCl permeation rate and KCl/$K_4Fe(CN)_6$ selectivity. Error bars of ion permeation rate are derived from three individual measurements based on three different samples. **h** Upper bound plot for ferro/ferricyanide permeability versus apparent ionic conductivity. Typical data reported previously in literature are included for comparisons such as those of commercial membranes including Nafion 212 and 115[31,33], Fumasep E610[12], Selemion CSO[43], and Celgard 3501[30], and those of recently reported ion-sieving membranes including porous PES/SPEEK[44], AO-PIM-1 and PIM-EA-TB[31], Graphene oxide[45], and SPX-BP-0.95 membranes[33]. The line represents an empirical upper bound for the trade-off between ionic conductivity and permeability of redox-active molecules for previously reported membranes. Solid symbols represent permeability data derived from dialysis diffusion tests and open symbols correspond to these derived from RFBs in operation. The values of IEC are labeled near the corresponding sPIM-SBF membrane symbols.

tortuous ion channels. Apparent ionic conductivity of sPIM-SBF membranes was measured with 1 M aqueous KCl (Fig. 3b), as an indication of the values for RFBs operated using the same supporting electrolyte. Apparent ionic conductivity is linearly correlated with sulfonate content of sPIM-SBF membranes with the

highest IEC value of 1.86 mmol g$^{-1}$ showing the highest ionic conductivity of $3.9 \times 10^{-2}$ S cm$^{-1}$ at 30 °C and $7.4 \times 10^{-2}$ S cm$^{-1}$ at 80 °C. The low activation energy for ion transport in sPIM-SBF-1.86 (0.07 eV, Supplementary Fig. 20) suggests thermally controlled ion dynamics in sPIM-SBF membranes where energy

barrier for ion conduction is reduced by accelerated segmental motion of polymer chains at high IEC upon swelling.

Intrinsic ionic conductivity of membranes was also measured in deionised water (Fig. 3c and Supplementary Fig. 21a), so as to decouple the two factors that dominate the mechanism of ion transport: the dissociation of potassium ions from sulfonate groups and the amount of adsorbed electrolytes in ion pathways within the membranes. sPIM-SBFs have an intrinsic ionic conductivity around half of their apparent ionic conductivity, with the percentage smoothly decreasing from 56% to 43% on increasing IEC (Supplementary Fig. 21b and Supplementary Table 8). This is attributed to the relative contribution to ion conduction from potassium ions dissociated from the sulfonate groups, in contrast to the adsorbed electrolyte, being more important for sPIM-SBFs with lower IEC. This agrees well with the electrolyte uptake capacity within sPIM-SBF membranes, which tend to swell more significantly and adsorb more electrolyte with higher content of sulfonate groups present in the sub-nanometer pores. For Nafion 115 membranes, the intrinsic ionic conductivity is only 27% of its apparent ionic conductivity, suggesting that ion conduction is dominated by absorbed electrolyte within their swollen, self-organized 1D channels[39], which gives poor size sieving selectivity as discussed below. Although high-free-volume PIM membranes generally undergo a rapid decrease of permeability due to the loss of free volume when applied to gas separation (known as physical ageing)[40], this phenomenon appears absent when the membranes are in fully hydrated state. Importantly, the ionic conductivity of all sPIM-SBF remains constant over an ageing period of around 400 days (Supplementary Fig. 22), indicating the long-term stability of ion exchange functionality and the retention of intrinsic porosity in supporting electrolytes.

**Selective transport of ions and redox molecules.** To investigate the selective ion transport in sPIM-SBF membranes, the permeability of salt ions and the crossover of redox-active electrolyte molecules were measured using concentration-driven dialysis diffusion tests (Fig. 3d and Supplementary Figs. 23, 24a). sPIM-SBF membranes achieve high permeability of small salt ions (e.g., KCl) with values up to $7.4 \times 10^{-6}$ cm$^2$ s$^{-1}$ for sPIM-SBF-1.86 (Supplementary Table 9), which are several orders of magnitude higher than charge-neutral hydrophilic PIM membranes (e.g., KCl permeability of $9.0 \times 10^{-9}$ cm$^2$ s$^{-1}$ for a 300 nm think PIM-EA-TB membrane[31]). The transport of K$^+$ ions through cation exchange membranes is promoted by the sulfonate groups fixed within membrane ion channels by electrostatic attraction. Despite the electroneutrality requirement that must be met in a concentration-driven diffusion leading to coupled transport of cations and anions, the transport of anions is controlled by multiple factors, particularly ion size, ion mobility and the membrane size sieving effect[41,42].

The permeability of a range of potassium salts was investigated to assess the selectivity of sPIM-SBF membranes towards anions with varied physical size and charge density, as the diffusion of the salts is predominantly controlled by the properties of the anions rather than that of the small K$^+$ cation of high mobility. sPIM-SBF membranes show a size-exclusion cut-off of ~7 Å for anions, allowing fast transport of small ions (e.g., Cl$^-$, NO$_3^-$) while rejecting larger ions (e.g., SO$_4^{2-}$, CO$_3^{2-}$, and Fe(CN)$_6^{4-}$) (Supplementary Figs. 24–25 and Supplementary Table 9). This correlates well with the pore size of sPIM-SBF (2~7 Å) measured from CO$_2$ gas sorption and molecular simulations. The ideal selectivity of Cl$^-$ (6.64 Å) over SO$_4^{2-}$ (7.58 Å) ranges from 550 for sPIM-SBF-0.53 to 30 for sPIM-SBF-1.86, much higher than that of the Nafion 115 membrane (selectivity of 5.4), suggesting

the potential use of ion-sieving membranes with well-defined subnanometer pores in applications requiring precise ionic separation such as water purification, sensor devices and a range of electrochemical devices.

The ferro/ferricyanide redox couple is widely used in the new generation of aqueous organic RFBs due to its low cost, fast redox kinetics, and reasonable redox potential[43], but these anions diffuse faster through membrane separators than organic redox-active molecules and cause unwanted crossover[31]. However, the concentration-driven diffusion rates of K$_4$Fe(CN)$_6$ through the sPIM-SBFs, for films with IEC values less than 1.40 mmol g$^{-1}$, measured using a dialysis diffusion cell over a 850 h testing period, are at least one order of magnitude lower than that of the Nafion membrane with roughly the same thickness (Fig. 3e and Supplementary Table 10). Similarly, diffusion rates of anionic organic molecules (i.e., 2,6-di(3-phosphenopropoxy)anthraquinone, 2,6-DPPAQ) are low with permeability in the order of $10^{-13}$~$10^{-12}$ cm$^2$ s$^{-1}$ (Fig. 3f), and the diffusion rates for the smaller 2,6-dihydroxyanthraquinone (DHAQ) are faster but the values are still comparable with Nafion and other ion-sieving membranes (Supplementary Figs. 26 and 27 and Supplementary Table 10). Indeed, such low permeability values of ferrocyanide and 2,6-DPPAQ can be considered as crossover-free and are mainly attributable to efficient size sieving through the rigid PIM backbones which restrict the thermal motions that result in the opening of a void with sufficient size for these large redox active species to move between free volume elements. At higher values of IEC (1.67 or 1.86 mmol g$^{-1}$), the sPIM-SBF membranes show ferrocyanide permeability similar to that of the Nafion membrane, presumably due to excessive swelling forming a pathway for larger anions. For sPIM-SBF membranes with IEC values less than 1.40 mmol g$^{-1}$, fast ion conduction and low permeability of redox-active materials result in their performance as a RBF membrane separator surpassing those of commercial cation exchange membranes[12,30,31,33,43] and recently reported ion-sieving membranes based on porous PES/SPEEK[44], hydrophilic PIMs[31,33], and graphene oxide[45] (Fig. 3g, h).

**Redox flow battery performance.** In contrast to conventional aqueous organic RFB chemistries operated with highly alkaline solutions (e.g., pH = 14)[6,9], electrolyte solutions with lower pH (i.e., pH range 9–12), which are commonly termed near-neutral pH in aqueous RFBs[12,13], are proven to be beneficial for minimizing the degradation of both redox-active molecules and polymer membranes[12,13,46]. Laboratory-scale asymmetric redox flow cells were constructed based on K$_4$Fe(CN)$_6$ (0.1 M) and 2,6-DPPAQ (0.1 M) operating at a pH of 9 (Supplementary Figs. 28 and 29) in an argon-filled glovebox for accelerated assessment of membrane performance. In these cells, membranes of sPIM-SBF generally show low area-specific-resistance (ASR) as measured by high-frequency electrochemical impedance spectroscopy (Fig. 4a). For example, the ASR of sPIM-SBF-1.86 membrane is as low as 0.82 Ω cm$^2$, which is significantly lower than that of Nafion 115 membrane (1.57 Ω cm$^2$) in an otherwise identical flow battery cell. The ASR values of sPIM-SBF membranes in RFBs agree well with their apparent ionic conductivity. Owing to their lower resistance, the cells based on sPIM-SBF membranes display enhanced energy efficiency and peak power density, relative to that of an otherwise identical RFB cell using Nafion 115 membrane (Fig. 4b, c and Supplementary Figs. 30–36). An RFB based on sPIM-SBF-1.40 membrane maintains high energy efficiency and a very low capacity decay rate of 0.0335% per day (0.0000795% per cycle) for about 120 h (2100 charge-discharge cycles), which is two orders of magnitude lower than that of an otherwise identical RFB using Nafion 115 membrane (2.17% per day and 0.00472% per

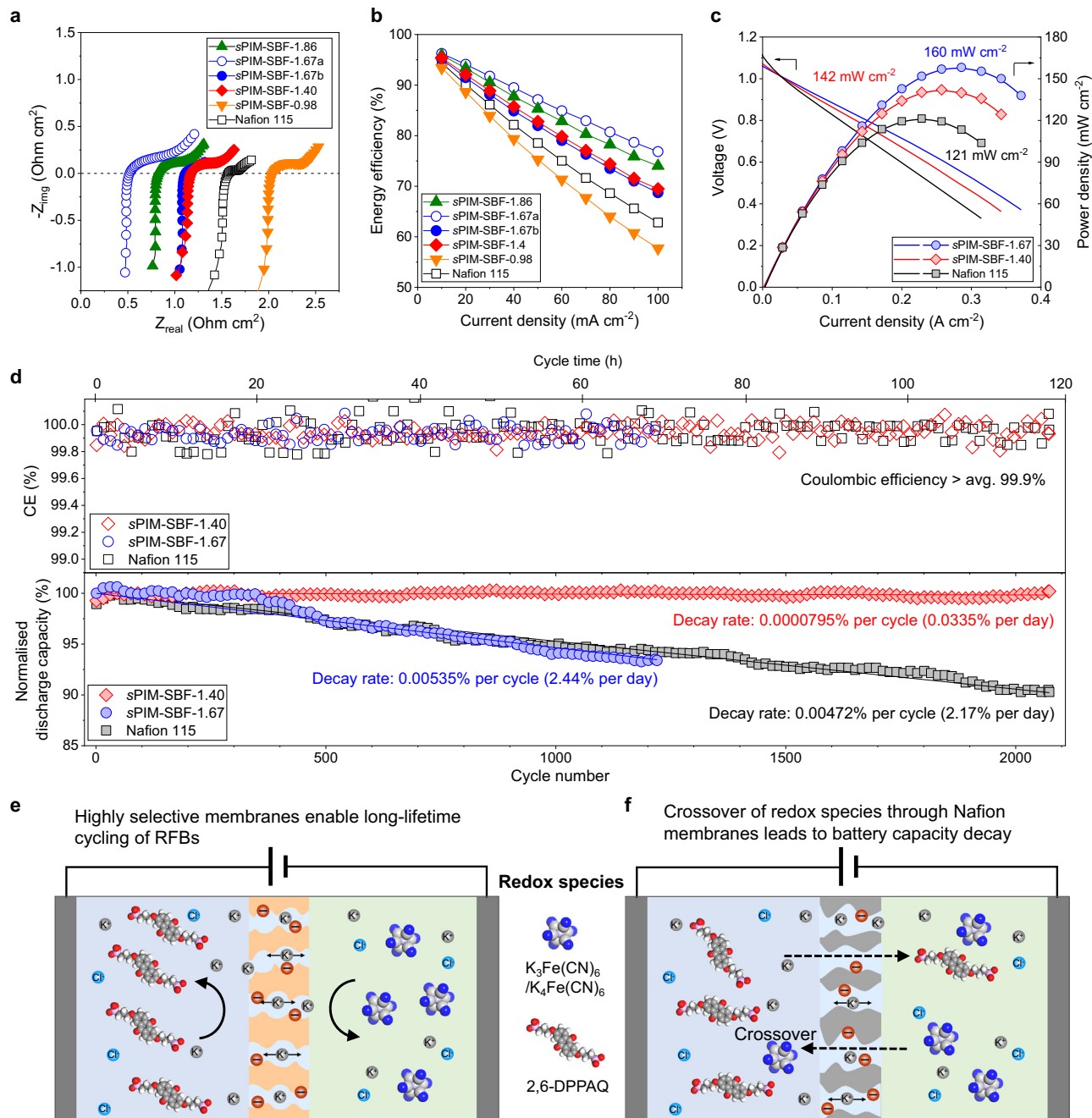

**Fig. 4 Low-concentration redox flow battery performance. a** EIS spectra of K$_4$Fe(CN)$_6$ (0.1 M)|2,6-DPPAQ (0.1 M) flow battery cells at 100% SOC.
**b** Energy efficiency as a function of current density in the range of 10–100 mA cm$^{-2}$. Membrane thicknesses are 142 μm (Nafion 115), 176 μm (sPIM-SBF-1.86), 45 μm (sPIM-SBF-1.67a), 177 μm (sPIM-SBF-1.67b), 134 μm (sPIM-SBF-1.40) and 139 μm (sPIM-SBF-0.98), respectively. **c** Voltage and power density versus current density at ~100% SOC. **d** Cycling stability of K$_4$Fe(CN)$_6$(0.1 M)|2,6-DPPAQ (0.1 M) RFBs assembled with sPIM-SBF-1.40 (134 μm), sPIM-SBF-1.67 (177 μm) and Nafion 115 membranes at pH 9 and a current density of 100 mA cm$^{-2}$. Normalized discharge capacity data are linearly fitted to derive capacity decay rates. The operating temperature in the glovebox is around 30 °C. Schematic illustrating the crossover mechanism that dominates battery capacity decay based on **e**, sPIM-SBF membranes and **f**, Nafion 115 membranes.

cycle) (Fig. 4d). An identical cell based on the less selective sPIM-SBF-1.67 membrane decays at a similar rate (0.00535% per cycle and 2.44% per day) to that of the cell with a Nafion 115 membrane, which is consistent with the time-dependent crossover rates measured from dialysis diffusion cells. It should be noted that these laboratory tests using low concentration of electrolyte resulted in multiple cycles each day, yet the results suggested that PIM membrane showed high rate of capacity retention due to minimal crossover over the testing period.

Performance decay of organic RFBs can be attributed to crossover of redox species, self-decomposition of active materials, or side reactions of electrolytes[47], or a combination of these problems. The decomposition and side reactions of redox active materials can be studied by advanced characterization techniques, such as in situ NMR and EPR techniques[48]. The analysis of the positive and negative electrolyte solutions after battery cycling, using cyclic voltammetry and ICP showed that the decomposition of these molecules in near neutral pH conditions (i.e., pH = 9) is

relatively slow. As visualized in Fig. 4e, f, the crossover of redox-active materials across the sPIM-SBF membranes leads to the loss of active materials in the capacity limiting side, and self-discharge due to the molecules diffused from the other excess side. Our results confirm that crossover is the predominant mechanism of performance decay for these aqueous organic RFBs operating at near-neutral pH (Supplementary Fig. 37). Therefore, we can conclude that the developed ion-selective membranes greatly improve the lifetime and stability of organic RFBs by limiting the crossover of redox-active species.

The use of high-concentration solutions of redox-active materials is vital to achieving high energy density RFBs. By mixing equal amounts of $K_4Fe(CN)_6$ and $Na_4Fe(CN)_6$, the solubility of ferrocyanide was increased to 1 M, higher than the 0.4 M concentration normally used in flow battery tests utilizing ferrocyanide redox couple[6,9,49]. For this high-concentration RFB, the reduced cell resistance as indicated by a low voltage gap of 0.27 V, attributable to the performance of the sPIM-SBF-1.40 membrane, enables an exceptionally high reversible capacity that is 93% of its theoretical value (Fig. 5a, b). Peak power density is significantly improved from 194 mW cm$^{-2}$ for Nafion 115 membrane to 303 mW cm$^{-2}$ for sPIM-SBF-1.40, and further to 374 mW cm$^{-2}$ for a less resistant sPIM-SBF-1.67 membrane (Fig. 5c and Supplementary Fig. 38), owing to the low internal resistance. This RFB exhibits a high and stable energy efficiency and a slow decay rate of 0.127% per day (0.00189% per cycle). In contrast, an otherwise identical RFB based on the higher resistance Nafion 115 membrane gives a higher voltage gap of 0.40 V, providing a capacity of only 88% of its theoretical value with a more rapid capacity loss of 2.66% per day (0.0355% per cycle) (Fig. 5d). Assuming that capacity loss solely comes from membrane crossover and RFB stacks are operated under ideal conditions (e.g., constant operating temperature, no oxygen penetration), these data allow the prediction that the highly ion-selective sPIM-SBF membrane would maintain 80% capacity performance of a practical RFB utilizing a high concentration of redox-active materials over 29 years, whereas, the same RFB using a Nafion 115 membrane would maintain 80% capacity performance for only 1.5 years (Supplementary Table 11 and Supplementary Note 1).

## Discussion

Low-cost membrane separators would have significant practical benefits for the widespread deployment of large-scale energy storage technologies. Although the multi-step laboratory-scale synthesis we propose would not be cost-effective as it stands, the procedure is scalable (e.g., there is no requirement for chromatographic separation of intermediates) and the cost per unit mass of the polymer would benefit greatly from large-scale production. Based on a simplified cost analysis (Supplementary Table 12), sPIM-SBF membranes are promising as a more economically competitive alternative to Nafion membranes. In addition, we anticipate that the sPIM-SBF would be used as the thin selective layer (<1 μm) on a cheap microporous support, therefore, the important cost per unit area of membrane will be reasonable. Our sulfonation method will also be applicable to other PIMs which will allow for optimization of materials for more economically competitive membranes. Furthermore, the ease of recovery of membrane components from end-of-life RFBs for subsequent uses constitutes a critical part in the life cycle assessment of a new membrane product. We characterized, by NMR, FT-IR, and GPC, sPIM-SBF membranes that were initially used in low-concentration RFBs for more than 2000 cycles (performance shown in Fig. 4) and then stored in supporting electrolyte solutions for around 20 months. No sign of degradation or structural change can be observed from these measurements (Supplementary Fig. 39). In addition, these membranes remain mechanically robust after being bent or folded repeatedly. Moreover, used sPIM-SBF membranes remain readily soluble in Dimethyl sulfoxide and can then be re-processed into robust free standing membranes (Supplementary Fig. 40). These results indicate that PIM membranes from spent RFB cells could either be directly re-used in a new battery cell or be re-dissolved to cast a fresh film.

In recent years, a wide range of flow battery chemistries have been developed beyond conventional vanadium flow battery, such as neutral or alkaline aqueous organic RFBs, semi-solid RFBs, metal-air RFBs, and non-aqueous RFBs[2–7]. It is challenging to find a one-size-fits-all solution and develop one specific type of membrane separators that finds universal applications in all types of flow battery systems. Membrane separators should be designed and tailored to meet specific requirements of each individual RFB chemistry. We envision that sPIM-SBF membranes are likely to be not sufficiently stable for long-term operation of vanadium flow batteries (e.g., cleavage of the dioxin-linkages and hydrolysis of nitrile groups), but we have demonstrated their improved ionic conductivity and selectivity towards organics, as well as stability and recyclability in near-neutral pH flow battery, which would advance the development and commercialization of aqueous organic flow batteries.

To allow comparison between the performance of sPIM-SBF membrane-based RFBs with those of recently reported aqueous organic RFBs, the components and key performance indicators are summarized in Supplementary Fig. 41 and Tables 13 and 14. Key parameters include membrane separators, electrolytes (types and concentration), supporting electrolytes (acidic, neutral, or alkaline), current density, energy efficiency, and capacity decay rate, and from these, it can be concluded that sPIM-SBF membranes significantly boost the energy efficiency and reduce the battery capacity decay rate under mild operation condition. It should be noted that some studies used organic redox molecules as capacity limiting side and excessive amount of $K_3Fe(CN)_6$/ $K_4Fe(CN)_6$ as the catholyte, therefore the decay rate reported in the literature could be lower than our measurements. Importantly, previous work on organic RFBs mainly focused on the molecular design of redox molecules while membranes were often both poorly understood and not optimized for the application. Our study proves that engineering the subnanometer pores of membranes and enhancing the membrane selectivity substantively improves the performance of this class of RFBs.

In summary, we have developed ion-sieving sulfonated polymer membranes that provide improved performance in aqueous organic RFBs operated at near neutral pH conditions (pH = 9). The rigid microporous structure enables the fast and selective transport of salt ions combined with effective prevention of crossover of the negatively charged redox-active species. The degree of sulfonation of the synthetic membranes can be tailored at a molecular level, delivering the necessary ion exchange capacity and appropriate channel size to achieve fast transport of charge-carrying ions while precisely size-excluding the redox-active species. By carefully combining these developed sulfonated PIM membranes with stable redox-active molecules and near-neutral pH electrolyte solutions, RFBs with high efficiency and significantly improved cycling stability were achieved.

## Methods

**Materials**. Nafion® 115 membrane (Dupont) was supplied from Sigma-Aldrich. Commercially available chemicals were used without further purification. All reactions using air/moisture sensitive reagents were performed in flame-dried or oven-dried apparatus under a nitrogen atmosphere.

**Synthesis of PIM-SBF**. The synthesis of 2,2′,3,3′-tetrahydroxy-9,9′-spirobifluorene (SBF) monomer precursors to PIM-SBF polymer is based on the published route[50] (Supplementary Fig. 2). SBF monomer (5.00 g, 13.14 mmol) and 2,3,5,6-

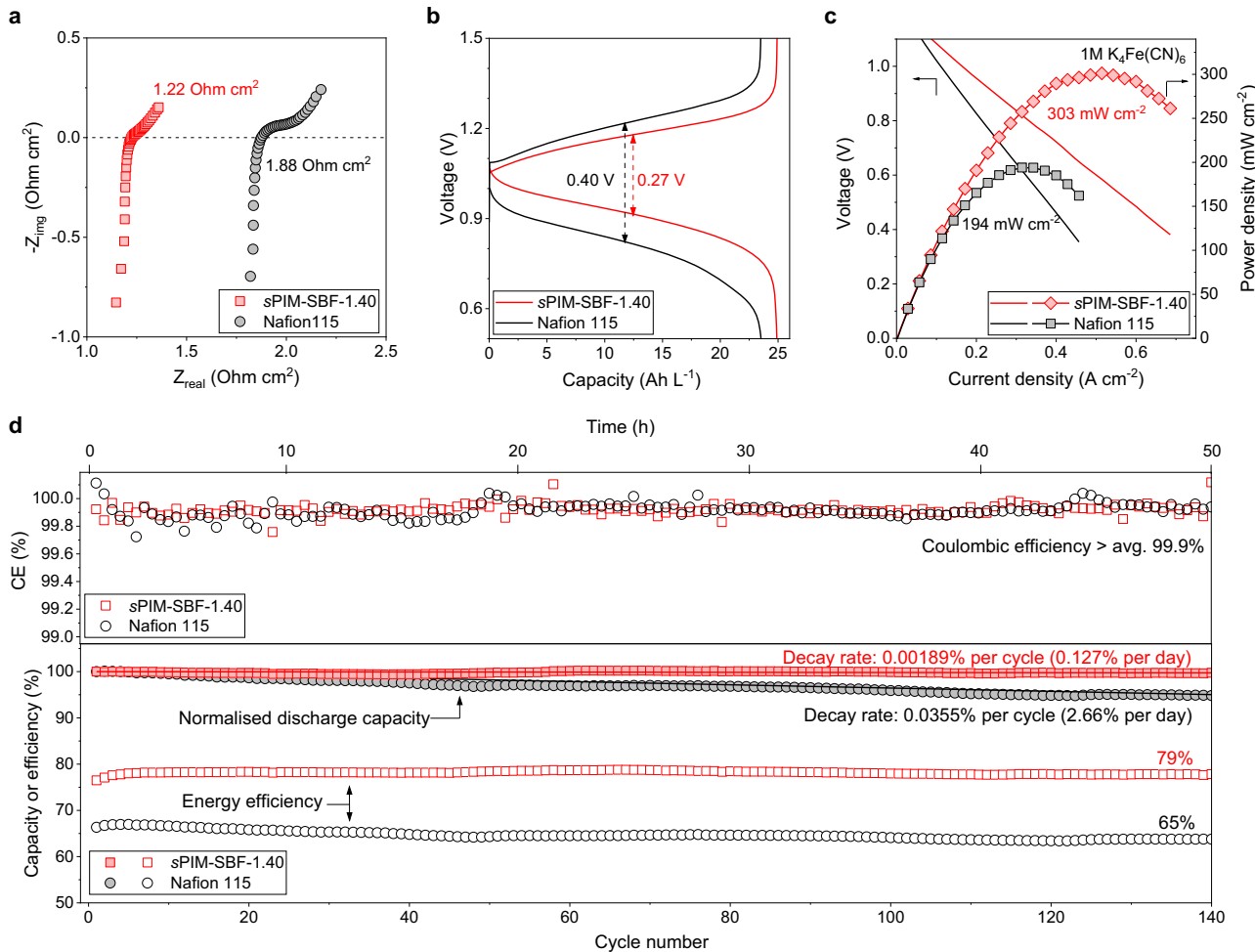

**Fig. 5 High-concentration redox flow battery performance. a** EIS spectra of $K_4Fe(CN)_6$ (1 M)|2,6-DPPAQ (0.4 M) flow battery cells at 100% SOC. **b** Capacity-voltage profile of the 2nd charging-discharging cycle at a current density of 100 mA cm$^{-2}$. **c** Voltage and power density versus current density at ~100% SOC using sPIM-SBF-1.40 and Nafion 115 membrane. **d** Cycling stability of RFBs using $K_4Fe(CN)_6$ (1 M)|2,6-DPPAQ (0.4 M) assembled with sPIM-SBF-1.40 or Nafion 115 membranes at pH = 9 and a current density of 100 mA cm$^{-2}$. Solid symbols represent normalized discharge capacity and open symbols represent energy efficiency. Normalized discharge capacity data are linearly fitted to derive capacity decay rates. The operating temperature in the glovebox is around 30 °C.

tetrafluoroterephthalonitrile (2.63 g, 13.14 mmol) were dissolved in anhydrous DMF (66 mL) at around 25 °C under a nitrogen atmosphere. Anhydrous $K_2CO_3$ (14.53 g, 105.15 mmol) was added and the mixture was heated at 65 °C for 92 h. On cooling, the mixture was poured into water (500 mL), acidified with hydrochloric acid (2 M) and filtered off. The precipitate was washed repeatedly with water, acetone, and methanol, reprecipitated twice from the mixture of acetone/methanol (2/1, v/v, 700 mL) to remove oligomers, and subsequently refluxed in methanol overnight to remove traces of residual solvent. The resulting product was collected by filtration and dried under *vacuum* at 110 °C for 12 h to afford the final polymer as yellow powder (5.59 g, 85%). $\nu_{max}$ (cm$^{-1}$): 3060, 2896, 2239, 1601, 1435, 1350, 1269, 1200, 1153, 1103, 1011, 868, 740; $^1$H NMR (500 MHz, CDCl$_3$): δ (ppm) 7.71 (br s, 2H, ArH), 7.41 (m, 4H, ArH), 7.13 (br s, 2H, ArH), 6.70 (br s, 2H, ArH), 6.38 (m, 2H, ArH); $^{13}$C NMR (126 MHz, CDCl$_3$): δ (ppm) 147.8, 145.2, 139.2, 124.0, 120.4, 112.7, 109.0, 108.4, 94.4, 65.4.

**Synthesis of sulfonated model compound (sSBF-OMe)**. A solution of TMSCS (230 mM) in CDCl$_3$ (100 μL) was added to a solution of 2,2′,3,3′-tetramethoxy-9,9′-spirobifluorene (SBF-OMe, 46 mM) in CDCl$_3$ (400 μL) at 273 K. The sample was then in situ monitored by $^1$H NMR over a period of 12 h at 278 K. **SBF-OMe:** $^1$H NMR (600 MHz,CDCl$_3$): δ (ppm) 7.71 (m, 2H, ArH), 7.34 (s, 2H, ArH), 7.32 (m, 2H, ArH), 7.03 (m, 2H, ArH), 6.66 (m, 2H, ArH), 6.23 (s, 2H, ArH), 4.03 (s, 6H, OCH$_3$), 3.64 (s, 6H, OCH$_3$); **sSBF-OMe:** $^1$H NMR (600 MHz,CDCl$_3$): δ (ppm) 7.89 (br m, 4H, ArH), 7.40 (br s, 2H, ArH), 7.10 (br m, 2H, ArH), 6.21 (br s, 2H, ArH), 4.03 (br s, 6H, OCH$_3$), 3.65 (br s, 6H, OCH$_3$).

**Synthesis of sPIM-SBFs**. The sulfonation of PIM-SBF polymer was achieved using trimethylsilyl chlorosulfonate (TMSCS) as the sulfonation agent as presented in

Supplementary Fig. 2. Detailed synthetic procedure of sPIM-SBF-1.40 was described below as an example. TMSCS (2.09 mL, 13.60 mmol, 1.7 equiv.) in anhydrous chloroform (5 mL) was added dropwise to a solution of PIM-SBF (4.0 g, 8.00 mmol, 1 equiv.) in anhydrous chloroform (135 mL) at 50 °C under a nitrogen atmosphere. The yellow reaction solution turned brown upon the addition of TMSCS, and was stirred at 50 °C for 24 h. On cooling, the brown reaction mixture was added to methanol and collected by filtration. The obtained solid was soaked in ammonium hydroxide (200 mL, 28% NH$_3$ in H$_2$O) at around 25 °C for 12 h to cleave the trimethylsilyl (TMS) groups. Subsequently, the solids were filtered off, stirred 3 times for 2 h with aqueous KCl solution (200 mL, 1 M), washed thoroughly with deionized water and finally dried under *vacuum* at 110 °C overnight to yield sPIM-SBF-0.98 in potassium ion form as an orange powder (4.68 g, 95%). sPIM-SBF polymers with five different degrees of sulfonation were prepared by adjusting the molar ratio of TMSCS and PIM-SBF repeating units to 5.0, 3.0, 1.7, 1.0, 0.5, and designated as sPIM-SBF-1.86, sPIM-SBF-1.67, sPIM-SBF-1.40, sPIM-SBF-0.98, sPIM-SBF-0.53 based on IEC, respectively. Since sPIM-SBF polymers with varied degree of sulfonation show same characteristic peaks in FT-IR and $^{13}$C NMR spectra, the FT-IR and $^{13}$C NMR characteristic peaks of sPIM-SBF-1.40 were listed below as an example. $\nu_{max}$ (cm$^{-1}$): 3400, 3060, 2239, 1601, 1435, 1350, 1269, 1180, 1103, 1030, 1011, 868, 740, 663, 623; $^{13}$C NMR (126 MHz, CDCl$_3$): δ (ppm) 148.3, 146.9, 138.6, 128.0, 125.7, 120.1, 111.5, 109.1, 93.0, 64.5.

**Synthesis of 2,6-di(3-phosphenopropyloxy)anthraquinone (2,6-DPPAQ)**. The redox-active molecule 2,6-DPPAQ was synthesized following a reported method[51]. A mixture of 2,6-dihydroxyanthraquinone (7.80 g, 32.47 mmol), diethyl (3-bromopropyl)phosphonate (21.00 g, 81.06 mmol), anhydrous $K_2CO_3$ (17.90 g, 129.51 mmol) and DMF (160 mL) was heated to and kept at 100 °C for 12 h. The mixture was cool down and DMF was removed. The solid was filtered off, washed

with water, and dried under suction to afford a pure ester precursor of 2,6-DPPAQ. Trimethylsilyl bromide (TMSBr, 49.71 g, 324.70 mmol) was added to a solution of the ester precursor in DCM (300 mL) at around 25 °C and stirred for 15 h. After distilling off DCM and excessive TMSBr, the crude precipitate was filtered, washed thoroughly with water and then hexane. The product was dried under *vacuum* to give a yellow powder (15.26 g, 97%). [1]H NMR (500 MHz, DMSO-$d_6$) $\delta$ (ppm) 8.10 (d, $J = 8.6$ Hz, 2H, ArH), 7.52 (d, $J = 2.7$ Hz, 2H, ArH), 7.37 (dd, $J = 8.7$, 2.7 Hz, 2H, ArH), 4.23 (t, $J = 6.4$ Hz, 4H), 1.97(m, 4H), 1.71 (m, 4H); [13]C NMR (126 MHz, DMSO-$d_6$) $\delta$ (ppm) 181.1, 163.4, 135.2, 129.5, 126.4, 120.6, 110.6, 68.5, 68.3, 24.4, 23.4, 22.8, 22.7 (Supplementary Fig. 28).

**Membrane formation**. Membrane formation was achieved by dissolving PIM-SBF polymer (0.6 g) in CHCl$_3$ (20 mL) or *s*PIM-SBF polymer (0.7 g) in DMSO (22 mL), which was then centrifuged (12,000 rpm for 5 min) to remove undissolved impurities and poured into a circular glass petri dish (diameter = 9 cm). Membranes were allowed to form by slow solvent evaporation in a desiccator for 48 h at a certain temperature (room temperature for PIM-SBF/CHCl$_3$ casting solution, and 65 °C for *s*PIM-SBF/DMSO casting solution). Free standing membranes with thickness around 150 μm were peeled off glass petri dishes and cut to needed sizes for further experiments.

**Characterization techniques**. [1]H and [13]C liquid state NMR spectra of original PIM-SBF, *s*PIM-SBF polymers, and 2,6-DPPAQ redox molecules were recorded in deuterated solvents using a Bruker AVA 500 spectrometer (500 and 125 MHz, respectively). PIM-SBF and 2,6-DPPAQ redox molecules were tested at 298 K with number of scans of 8 and 128 for [1]H and [13]C NMR. Specifically, high-concentration solution samples of *s*PIM-SBF polymers (50 mg) in DMSO-$d_6$ (0.75 mL) were prepared and tested at 333 K. [1]H and [13]C spectra were typically compiled from 32 and 4096 scans, respectively. Sulfonation reaction of model compound was monitored by [1]H liquid state NMR at 278 K using a Bruker AVA 600 instrument to verify the possible sulfonation sites in PIM-SBF polymer. Infrared spectra were recorded in the range 4000–500 cm$^{-1}$ using a Perkin-Elmer 1600 FTIR spectrometer with polymer powder or membrane samples mounted on a zinc-selenium/diamond plate. All absorptions are quoted in cm$^{-1}$. GPC measurement of original PIM-SBF was carried out by a GPC MAX 1000 system equipped with two Viscotek columns (CLM3012 LT 5000 L) and a RI detector (VE3580) using CHCl$_3$ (~1 mg mL$^{-1}$) at a flow rate of 1 mL min$^{-1}$ as an eluent. GPC traces of *s*PIM-SBF polymers were obtained by a GPC Agilent 1260 system equipped with two PLgel MIXED-C columns (200-2,000,000 g mol$^{-1}$, 5 μm) and a RI detector using DMF containing 0.1% w/v lithium bromide (~5 mg mL$^{-1}$) at 60 °C at 1 mL min$^{-1}$ as an eluent. Calibration was performed using narrow dispersity poly(methyl methacrylate) standards. Tensile tests on membranes were measured on a Lloyd-Ametek EZ50 Material Testing Machine under 50% relative humidity (RH) and at room temperature. Specimens with a width of 5 mm and an effective length of 30 mm were tested at a strain rate of 3 mm min$^{-1}$ (10% min$^{-1}$). Average values and standard deviation of the Young's modulus were calculated on 3–4 specimens. The TGA was performed using a NETZSCH STA 449 F5 Jupiter thermogravimetric analyser. Polymer samples were heated at a heating rate of 10 °C min$^{-1}$ from room temperature to 1000 °C under a nitrogen atmosphere. Skeletal density tests of polymer powders were conducted on a Micromeritics Accupyc II 1340 helium pycnometer including a 3.5 cm$^3$ sample chamber at 25 °C. Samples were degassed for 12 h at 110 °C under *vacuum* prior to measurement. The mean values and standard deviation of skeletal densities were determined on a series of 10 measurements. Low-pressure gas physisorption measurements of PIM powders were carried out on a Micromeritics 3Flex surface characterization analyser. Samples were degassed at 110 °C under high *vacuum* for 12 h, and loaded into the instrument and in situ degassed at 110 °C for another 12 h prior to analysis. N$_2$ adsorption/desorption isotherms were measured at 77 K, and CO$_2$ adsorption/desorption isotherms measured at 273 K. Dynamic water vapor sorption was measured on a dynamic vapor sorption (DVS) Endeavor gravimetric sorption analyser (Surface Measurement Systems). Membrane samples (20–30 mg, around 150 μm thick) were dried at 110 °C under *vacuum* for 12 h, and loaded into the apparatus and in situ dried at 25 °C under flowing dry air for at least 24 h until the mass became constant. Water sorption behavior was obtained by measuring the change in mass of each sample from dry to fully hydrated state at a specified relative humidity.

**Ion exchange capacity (IEC) determination**. The IECs (mequiv/g) of *s*PIM-SBF membranes were determined by the acid-base titration method[33]. Membranes were soaked in HCl solution (0.1 M) for 6 h (repeat four times) at room temperature to protonate all sulfonate groups. Protonated membranes were rinsed and immersed in deionized water for 48 h to remove residual HCl. Dry *s*PIM-SBF membranes in H$^+$ form were immersed in NaCl solution (1 M) for 6 h (repeat four times) to convert H$^+$ form into Na$^+$ form. The exchanged H$^+$ was then titrated with NaOH solution (0.01 M, standardized by 0.010 M potassium hydrogen phthalate) using the phenolphthalein indicator. The values of IEC were determined by the Eq. (1):

$$IEC = \frac{C_{NaOH} V_{NaOH}}{W_d} \tag{1}$$

where $C_{NaOH}$ is the concentration of NaOH solution, $V_{NaOH}$ is the volume of NaOH solution, and $W_d$ is the weight of dry *s*PIM-SBF membranes in H$^+$ form. Error bars are standard deviations derived from 3 IEC measurements based on three different samples.

The water content $\lambda$ (i.e., the number of water molecules per sulfonate group) for sulfonated membranes can be obtained using dynamic water vapor sorption and IEC data[52] using Eq. (2):

$$\lambda = \frac{n(H_2O)}{n(SO_3^-)} = \left(\frac{\Delta M_w}{\overline{M}_{H_2O}}\right) \times \frac{1000}{IEC} \tag{2}$$

where $\triangle M_w$ is the weight of water absorbed by membranes samples at target humidity from DVS, $\overline{M}_{H_2O}$ is the molar mass of water in 18 g mol$^{-1}$.

**Water/electrolyte uptake (WU/EU) measurement**. Membrane specimens in potassium ion form were dried at 110 °C under *vacuum* for 12 h to obtain the dry mass. These specimens were immersed in deionized water or KCl electrolyte (1 M) for 24 h at around 25 °C. The wet mass of membranes was measured immediately after wiping off excess liquid from the surface. The water or KCl electrolyte uptake was given by Eq. (3):

$$WU/EU(\%) = \frac{W_{wet} - W_{dry}}{W_{dry}} \times 100\% \tag{3}$$

where $W_{wet}$ and $W_{dry}$ are fully hydrated mass and dry mass of membrane specimens, respectively. The mean values and error bars are derived from three measurements based on three different membranes.

**Swelling ratio (SR) measurement**. The swelling ratio was determined from the difference in linear dimensions between wet ($x_{wet}$) and dry ($x_{dry}$) membranes in potassium ion form, and as calculated according to Eq. (4):

$$SR(\%) = \frac{x_{wet} - x_{dry}}{x_{dry}} \times 100\% \tag{4}$$

Error bars are standard deviations derived from 3 SR measurements based on three different samples.

**Ionic conductivity measurement**. Ionic conductivity of membranes was measured by 2-electrode electrochemical impedance spectroscopy (EIS) using the potentiostat mode at an AC bias of 10 mV and a frequency range from 0.5 MHz to 10 Hz with 20 data points collected per decade. Membrane samples in K$^+$ form were soaked in 1 M aqueous KCl (for apparent K$^+$ conductivity test), or deionized water (for intrinsic K$^+$ conductivity test) for 24 h, then sandwiched between two stainless steel electrodes and sealed with a coin cell (Type 2032). For intrinsic H$^+$ conductivity test, membrane samples in K$^+$ form were immersed in HCl solution (0.1 M) for 6 h (repeat four times) to convert K$^+$ form into H$^+$ form, then soaked in deionized water for 48 h to remove residual HCl before cell assembly. The assembly procedure was performed in KCl electrolyte or deionized water to avoid air bubbles being trapped in cells. The membrane conductivity over a temperature range from 30 to 80 °C was calculated from the resistance using the following Eq. (5)[31]:

$$\sigma = \frac{L}{AR_m} \tag{5}$$

where $\sigma$ is the ionic conductivity of membranes in S cm$^{-1}$, $L$ is the membrane thickness in cm, $A$ is the effective membrane area of 2.00 cm$^2$, $R_m$ is the membrane resistance in Ω. $R_m$ is obtained by subtracting the resistance of the assembled coin cell with that of an otherwise-identical cell without any membrane.

**Diffusion and Crossover measurement**. Ion diffusion dialysis and redox molecule crossover tests were performed using stirred H-shaped cells. Membranes were sandwiched between two polydimethylsiloxane O-rings and then sealed in the middle of H-shaped cells by clips. Magnetic stirring was applied in both feed and permeate solution in order to alleviate concentration polarization near membranes. In ion diffusion dialysis tests, salt aqueous solution (0.4 M KCl, 0.4 M KNO$_3$, 0.2 M K$_2$SO$_4$, and 0.2 M K$_2$CO$_3$, 50 mL) was used as a feed solution and deionized water (50 mL) was used in the permeate side. The ionic conductivity of the permeate side was continuously recorded by a conductivity meter (Orion Star A210, Thermo Scientific) at the time interval of 30 s. The concentration change of salt solution over time was calculated from the corresponding salt calibration curves based on the linear relationship between the ionic conductivity and concentration of salt aqueous solution. The mean values and error bars of ion permeation rate were derived from three individual measurements based on three different samples.

In redox molecule crossover tests, K$_4$Fe(CN)$_6$ (0.1 M) or 2,6-DPPAQ (0.1 M) in KCl solution (1 M, 50 mL, pH = 9.0) was used as a feed solution and blank KCl aqueous solution (1 M, 50 mL, pH = 9.0) was used in the permeate side. 2,6-DHAQ (0.1 M) in KOH aqueous solution (1 M, 50 mL) was used as a feed solution and blank KOH aqueous solution (1 M, 50 mL) was used in the permeate side. The concentration change of K$_4$Fe(CN)$_6$ in the permeate solution was quantitatively detected by inductively coupled plasma-optical emission spectrometry (ICP-OES). Permeate aliquots were collected from the permeate side three times per day, and

diluted for 100 times with 2.0 wt.% $HNO_3$ prior to analysis. The concentration change of 2,6-DPPAQ or 2,6-DHAQ in the permeate solution was quantitatively detected by the calibrated UV-Vis spectrometer. Permeate aliquots were collected from the permeate side once per week, analyzed without dilution, and recycled to permeate side.

In above concentration-driven diffusion and crossover experiments, the permeation rate (flux) of ions and redox molecules across the membrane over a short-time period without volume change follows Fick's first law:

$$J = \frac{V}{A}\left(\frac{\partial C}{\partial t}\right) \quad (6)$$

where $J$ is the permeation rate in mol $cm^{-2}\,s^{-1}$, $V$ is the solution volume of 50 mL, $A$ is the effective membrane area of 1.54 $cm^2$, $C$ is concentration of the permeate solution in mol $cm^{-3}$, and $t$ is diffusion time in s. During the process, negligible change for the difference between permeate and feed concentration can be ignored (that is, $C_1 - C_2 = C_1$), and the flux can be assumed to be constant. Consequently, Fick's first law can be simplified as Eq. (7):

$$J = \frac{P(C_1 - C_2)}{l} = \frac{PC_1}{l} \quad (7)$$

where $P$ is the permeability in $cm^2\,s^{-1}$, $C_1$ is concentration of the feed solution in mol $cm^{-3}$, and $l$ is the thickness of membranes in cm.

**Cyclic voltammetry.** Cyclic voltammetry measurements were performed using a three-electrode configuration composed of a glassy carbon working electrode, a Pt counter electrode and an Ag/AgCl reference electrode at around 25 °C. The electrolytes were prepared by dissolving redox-active materials (1 mmol) in aqueous KCl solution (1 M, 10 mL), using trace amount of aqueous KOH solution (1 M) to adjust pH to 9.0) Cyclic voltammogram of $K_4Fe(CN)_6$ and 2,6-DPPAQ were collected using a Biologic SP-150 potentiostat at a scan rate of 100 mV $s^{-1}$ as shown in Supplementary Fig. 29. The crossover rates of 2,6-DPPAQ through membrane separators to catholytes, and $K_4Fe(CN)_6$ to analytes in the operating battery were measured by cyclic voltammetry.

**Flow battery tests.** A cell hardware (Scribner Associates) with POCO single serpentine pattern graphite plates was used to assemble the flow cells. A piece of ~150-μm-thick membrane was sandwiched between electrodes (effective geometric area = 7 $cm^2$) comprising a stack of three sheets of carbon paper (SGL), with the rest of the space between graphite plates sealed by Viton gasket. Electrolytes were fed into the cell at a flow rate of 100 mL $min^{-1}$ through a Cole-Parmer peristaltic pump. The flow cell measurements were constructed based on $K_4Fe(CN)_6$ (0.1 M) and 2,6-DPPAQ (0.1 M) operating at a pH of 9 (Supplementary Figs. 28 and 29) in an argon-filled glovebox.

Carbon papers were pretreated by baking at 400 °C in air for 24 h. Nafion membranes were pretreated by soaking in 80 °C deionized water for 20 min, and then in 25 °C hydrogen peroxide solution (6%) for 35 min, before storing in aqueous KCl solution (0.1 M) at around 25 °C. Prior to full cell tests, Nafion and sPIM-SBF membranes were soaked in 1 M aqueous KCl (for 0.1 M flow cell tests) or a mixed electrolyte solution of 2.5 M NaCl and 2.5 M KCl (for 1 M flow cell tests) for 24 h.

For 0.1 M flow cells, the electrolytes consisted of 0.1 M $K_4Fe(CN)_6$, or 0.1 M 2,6-DPPAQ combined with 0.4 M KOH in 10 mL 1 M aqueous KCl solution. For 1 M flow cells, catholyte was prepared by dissolving 2.5 mmol $K_4Fe(CN)_6$ and 2.5 mmol $Na_4Fe(CN)_6$ in 5 mL mixed supporting electrolyte of 0.5 M KCl and 0.5 M NaCl, resulting in 1 M $[Fe(CN)_6]^{4-}$, 1 M $Cl^-$, 2.5 M $K^+$ and 2.5 M $Na^+$. To make the cation concentration equivalent in both electrolytes, 5.0 mmol 2,6-DPPAQ and 20.0 mmol KOH were dissolved in 12.5 mL mixed supporting electrolyte of 0.9 M KCl and 2.5 M NaCl, producing anolyte containing 0.4 M deprotonated 2,6-DPPAQ, 3.4 M $Cl^-$, 2.5 M $K^+$ and 2.5 M $Na^+$. For both 0.1 M and 1 M flow cells, trace amount of KOH was added to catholyte and analyte solutions to adjust the pH to 9.0.

To obtain an electrochemical polarization curve, the cell was charged to a desired state of charge (SOC), and then polarized using linear galvanic sweep method at a rate of 200 mA $s^{-1}$ from −6000 to 6000 mA. The corresponding power density at specific SOC (20, 50, ~100 %) was derived from the current–voltage curve.

The flow cell stability tests were operated in an argon-filled glovebox (Oxygen < 2 ppm, water < 2 ppm, temperature around 30 °C). RFBs are cycled at current density ranging from 10 to 100 mA $cm^{-2}$ in the first 110 cycles, before cycling at a constant current density of 100 mA $cm^{-2}$. Cycling data is presented from the 111st cycle, which is labeled as the 1st cycle here and the data is shown for every 20 cycles.

**Computational methods.** Molecular dynamics simulations were conducted using the Large-scale Atomic/Molecular Massively Parallel Simulator LAMMPS software package. Interactions were described by the all-atom optimized potentials for liquid simulations (OPLS-AA) with additional parameters for the sulfonic acid groups[53,54]. 1.14*CM1A-LBCC partial charges for the polymer were calculated using LigParGen[55]. Sulfonic acid pendant groups were described in their

protonated state to represent the dry polymer structure. Five independent models were generated for each system.

Polymer chains of PIM-SBF and sPIM-SBF analogues were constructed using the previously published Polymatic simulated polymerization algorithm and 21-step equilibration protocol[56]. The Polymatic procedure consists of five steps: (1) 2 repeat units of PIM-SBF and sPIM-SBF were first optimized to the B3LYP/6-31 G** level in Gaussian16[57]. LAMMPS topology files for the individual monomer units, sulfonated and non-sulfonated, were then generated from the optimized geometries. Reactive atoms were defined where the polymerization algorithm would later form new bonds. (2) 150 PIM-SBF/sPIM-SBF monomers were randomly packed into a periodic box at an arbitrary low density (~0.36 g $cm^{-3}$). The ratio of PIM-SBF to sPIM-SBF monomers was adjusted accordingly to mirror the IECs of the experimental materials (0, 0.53, 0.98 and 1.40, 1.67, and 1.86 g $mmol^{-1}$). (3) Monomer units were 'polymerized' via the Polymatic simulated polymerization algorithm to achieve long chain lengths. During the polymerization, artificial charges were placed on reactive atoms ($q_{polym} = \pm\,0.3e$) to increase bias towards bond formation. During the polymerization step, new bonds are formed between reactive atoms on neighboring monomers within a cut off distance (6 Å). Artificial charges were removed after bond formation. (4) Any unreacted chain ends were saturated with a capping group to mirror the experimental precursors. (5) Finally, the polymer chains were annealed by means of a 21-step molecular dynamics equilibration of gradual compressions and decompressions set out by Colina and co-workers[56]. This procedure is well established in the literature for the simulation of microporous structures with experimentally comparable densities. The parameters were set as recommended in the literature[56], where $T_{final}$ = 300 K, $T_{max}$ = 1000 K, $P_{final}$ = 1 bar and $P_{max}$ = 5 ×$10^4$ bar. Supplementary Table 4 reports the IEC, degree of sulfonation, initial box length, degree of polymerization and the density before and after the amorphous structure generation procedure for each system.

The pore networks of equilibrated systems were characterized by Zeo + +[58]. 5000 Monte Carlo (MC) samples for surface area calculations, and 50,000 MC samples for pore volume, pore size distributions and visual pore size distributions were found to be adequate to balance computational cost with accuracy (results within <1% of those at the highest levels of sampling). The high accuracy setting was used for all of the calculations. The results shown for each system are the average of the 5 independent samples.

A sPIM-SBF polymer chain of length $n$ = 2 and $SO_3^-$ per monomer unit = 1 was optimized in Gaussian16[57] using the B3LYP density functional coupled with 6–31 + G** basis set. The optimal geometry was confirmed by a frequency calculation to ensure that it was a true minimum. The electrostatic potential surface (ESP) of the polymer was visualized in Gaussview on an isosurface of electronic density 0.001 a.u.

**Reporting summary.** Further information on experimental design is available in the Nature Research Reporting Summary linked to this paper.

## Data availability
The authors declare that all the data supporting the findings of this study are available within the paper and its Supplementary Information files or from the corresponding author upon request.

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

## Acknowledgements

This project has received funding from the European Research Council (ERC) under the European Union's Horizon 2020 research and innovation program (grant agreement No 851272, ERC-StG-PE8-NanoMMES) and (grant agreement No 758370, CoMMaD). This work was also funded by the Engineering and Physical Sciences Research Council (EPSRC, UK, EP/M01486X/1, EP/V047078/1, EP/P024807/1, EP/S032622/1), and EPSRC Center for Advanced Materials for Integrated Energy Systems (CAM-IES, EP/P007767/1) and Energy SuperStore (UK Energy Storage Research Hub). C.Y. acknowledges a full PhD scholarship funded by the China Scholarships Council/University of Edinburgh. A.W. acknowledges a full PhD scholarship funded by the Department of Chemical Engineering at Imperial College. R.T. acknowledges a full PhD scholarship funded by the China Scholarship Council. C.Y. and A.W. acknowledge the Royal Society of Chemistry Researcher Mobility Grant. C.B. acknowledges the EPSRC ICASE PhD studentship funded by EPSRC and Shell. K.E.J. acknowledges the Royal Society University Research Fellowship. The authors acknowledge Juraj Bella for help with NMR measurements, and Meltem Haktaniyan for help with GPC measurements.

## Author contributions

Q.S. and N.B.M. conceived and supervised the project. C.Y. synthesized and characterized PIM-SBF, sPIM-SBF and 2,6-DPPAQ, and prepared and characterized membranes. A.W. carried out characterizations of sPIM-SBF and membranes, and performed RFB tests. C.G.B. helped with PIM-SBF synthesis. R.T. helped with RFB tests. C.B. and K.E.J. contributed to the molecular simulation and analysis. E.H.-S. and D.R.W. contributed to DVS measurements. N.P.B. and A.R.K. provided facility support and new insights into the research. P.A.A.K. contributed to industrial application perspective. Q.S., N.B.M., C.Y., and A.W. wrote the manuscript with contributions from all authors. All authors discussed the results and commented on the manuscript at all stages.

## Competing interests

The authors declare no competing interests.
