## [Peer Review File · Nature Communications]

Development of efficient aqueous organic redox flow batteries using ion-sieving sulfonated polymer membranesReviewers' Comments:

Reviewer #1:

Remarks to the Author:

Report Nat. Comm.

The work by Neil B. McKeown, Qilei Song et al constitutes an interesting study addressing a very important topic of our days: storage of energy. The work is basically totally devoted to the use of a spirobifluorene based polymeric membrane as an efficient substituent for the conventional nafion membrane. Whereas the topic is of relevance since the storage of energy, in particular with RFBs, is nowadays a very important topic driven by the Climate Change Agenda in the search for sources of clean energy. It would be worth to complement with some additional work. Otherwise it would be more likely to be found in a more specific journal as J. Membrane Science.

The work is almost totally devoted to the study of a spirobifluorene based polymer of intrinsic microporosity (PIM-SBF) membrane to replace the conventional nafion.

1) It would be important to estimate the cost of production (with synthesis included) of this polymer so that it can be compared with nafion. This can be done and added to SI.

2) one of the advantages of the nafion membrane is the easiness of its full recover for subsequent uses. The indication that the PIM membranes afford stable battery operations over 2000 charge and discharge cycles does not guarantee that the membranes are recoverable. This should be discussed.

3) The sulfonated polymers are well characterized, no doubt. However, characterization of the 2,6-DPPAQ along the cycles (beginning, middle and final) would be helpful to the interpretation of the viability of the RFB working with the PIM-SBF membrane. This is partially done with data in Supplementary Fig. 26, but not under working conditions. It is also specifically mentioned on page 12: "The analysis of the positive and negative electrolyte solutions after battery cycling, using NMR spectroscopy, cyclic voltammetry, and ICP showed that the decomposition of these molecules in near neutral pH conditions is relatively slow.", but then it reports to fig 4 e and f which are schematically and to Supplementary Fig. 36 that shows the CV profiles of redox active molecules after RFB full cell cycling tests. These seem to show some level of changes but are not directly compared to the initial CVs.

4) Being the work essentially focused in a new membrane, it would be advisable to make the comparison with the nafion membrane with a vanadium RFB, in addition to the 2,6-DPPAQ, $K_4Fe(CN)_6$ pair.

5) Indication that pH9 is near neutral pH is erroneous and should be avoided. Neutral pH is 7 -7.5; 9 is an alkaline pH value.

Other issues:

BET (Brunauer–Emmett–Teller) surface should not always be abbreviated

In the Supplementary information section when the battery lifetime is predicted an energy storage capacity value is given of 2kWh for an active surface area of $1m^2$. How the value of energy storage capacity is obtained should be better elucidated.

Reviewer #2:

Remarks to the Author:

The authors report on a nano-porous ion-selective membrane for redox flow batteries using large organic molecules as the active species, operating with different electrolytes on either side of the

membrane. They claim low crossover of active species through size exclusion as well as Donnan exclusion. They report characterization data of the membrane as well as data obtained by cycling a flow battery.

The polymer chemistry reported as well as the polymer characterization is sound. However, there are several aspects of this paper that detract from its publication in Nature Communications:

1. The reported conductivities are quite low in spite of measurement in KCl solution.
2. Most data is presented without any error bars (something which absolutely needs to be remedied)
3. The conclusion of Donnan exclusion coupled with size-exclusion does not appear to be supported by evidence. The effect appears to be purely that of size exclusion, which is well known and has been previously reported.
4. While the cycling numbers seem to be impressive, they also are more a bit disingenuous - the cycle duration seems to be extremely low - perhaps only minutes - while multiple hours / cycle would be the norm. Unless data can be presented at this scale, not many conclusions about suitability can be drawn.

For these reasons, it would be difficult to recommend in favor of publication.

Reviewer #3:

Remarks to the Author:

This manuscript reports a new membrane consisting of spirobifluorene-based polymers with intrinsic microporosity (PIM-SBF) for aqueous organic redox flow batteries (AORFB). By modulating the sulfonation degree of the polymers, the authors attempted to control their porosity and ion conductivity as well as ion selectivity in the crossover. The new membrane fabricated by one polymer (sPIB-SBF-1.40) showed higher ionic conductivity, KCl permeation rate, and anion selectivity than commercial Nafion 115 membrane, which finally led to higher cycle stability in AORFB using 2,6-DPPAQ and $K_4Fe(CN)_6$.

This work is well written, and the authors systematically investigated the structure-property relationship between sulfonation degree of the polymers, their physicochemical properties, and membrane characteristics. However, a few issues should be addressed before acceptance.

1. The molecular weights and the PDI values of the polymers used in this study should be reported.
2. The authors characterized the chemical structure of the sulfonated polymers using solid-state ^{13}C NMR. But, sPIM-SBF polymer membranes were fabricated from DMSO solutions. If the polymers were soluble, why did they characterize the polymers in solid-state? If possible, solution-state NMR characterization should be performed. In my opinion, it can provide a more accurate and quantitative characterization for the sulfonation.
3. The raw data for all NMR measurements should be provided in the supplementary information.
4. In this study, it was found that the sulfonation clearly decreased the BET surface area and pore volume of the polymers. However, as the sulfonation degree increased, the ion permeation rate also increased (Fig 3). It is surprising that the increasing rate is steeper in the case of larger anions (Table S8). Furthermore, the ion selectivity was lowered in the more sulfonated polymers (Fig 3 and Table S9), despite lower BET surface, pore-volume, and highly negatively charged backbone structures. The authors should provide a more detailed and reasonable explanation for these behaviors.

Response to reviewers' comments

We are grateful to all reviewers for their time and constructive comments. Below are the point-to-point response.

Reviewers' comments (Blue); Authors' response: black; Revised text in manuscript and ESI (Red).

Reviewer #1 (Remarks to the Author):

The work by Neil B. McKeown, Qilei Song et al constitutes an interesting study addressing a very important topic of our days: storage of energy. The work is basically totally devoted to the use of a spirobifluorene based polymeric membrane as an efficient substituent for the conventional Nafion membrane. Whereas the topic is of relevance since the storage of energy, in particular with RFBs, is nowadays a very important topic driven by the Climate Change Agenda in the search for sources of clean energy. It would be worth to complement with some additional work. Otherwise it would be more likely to be found in a more specific journal as J. Membrane Science. The work is almost totally devoted to the study of a spirobifluorene based polymer of intrinsic microporosity (PIM-SBF) membrane to replace the conventional Nafion.

Response: We thank the reviewer for offering valuable comments and suggestions. As the reviewer pointed out, we believe our work will contribute substantially to the development of new ion-conductive membranes and advance their applications in a range of energy and environmental processes, particularly those in the renewable energy conversion and storage including redox flow batteries (RFBs). The reviewer had concerns about the cost of production, recovery and reuse and the universality of our membranes in different RFB chemistries. However, we need to explain that the novelty of this work does not lie in the synthetic chemistry itself or membrane manufacturing. The design strategy of making ion-conductive membranes that combine fast ion transport with precise molecular sieving functions enable long-life organic RFBs under mild and environmental-friendly operation conditions, based on the sulfonation of a spirobifluorene based polymer of intrinsic microporosity (sPIM-SBF). **sPIM-SBF is the prototypical PIM-based polymer demonstrating its exceptional performance as an RFB membrane separator.** Furthermore, we and other research groups inspired by this work will make cheaper and simpler sulfonated PIM-based polymers for membrane process to match the scales required by the commercial and residential deployment of RFBs.

Detailed explanations and additional experiments have been given to address all the major and minor issues raised by the reviewer. We hope the reviewer will value our efforts and recognize the significance of our work in a broad context of a new generation of ion-conductive PIM membranes and the broad impact on energy conversion and storage.

1) It would be important to estimate the **cost of production** (with synthesis included) of this polymer so that it can be compared with Nafion. This can be done and added to SI.

Response: The cost of polymer synthesis and membrane manufacturing will become a critical issue when we scale up the membranes for large scale applications. The primary goal of our work is to demonstrate the concept of sulfonated PIM membranes to achieve fast and selective ion transport in redox flow batteries. At this early stage, we demonstrated the concept in laboratory scale and our polymer design is still at low TRL level. Further work will be required to scale up the synthesis or developing alternative synthetic chemistry using low-cost monomers. Nevertheless, the production cost of sPIM-SBF-1.40 was estimated as an example based on the cost of raw materials, reagents and catalysts used in the synthesis (cost data were accessed on websites of common chemical vendors on 01 Dec. 2021) as shown in Supplementary Table 13. The final cost would be £ 2.4 per gram for sPIM-SBF polymers and £ 120 per m² for sPIM-SBF membranes (50 µm thick). It should be noted that this simple estimation does not include the costs associated with manufacturing and operational costs. We envision that large scale synthesis and manufacturing will further reduce the cost of these polymer membranes.

Furthermore, thanks to the solution processability, PIM polymers can be processed into thin-film composite membranes with a sub-micron layer of sPIM-SBF supported on a low-cost substrate, through well-established manufacturing techniques such as roll-to-roll casting (Jue, M. L. & Lively,

R. P. *Curr. Opin. Chem. Eng.* **2022**, 35, 100750). Several m² of thin film composite membranes could be prepared from 1 g of polymer, hence reducing the total cost of the composite membranes. Our group has demonstrated the feasibility of making PIM-based thin-film composite membranes for redox flow batteries (Tan, R. *et al. Nat. Mater.* **2020**, 19, 195-202).

Recently, Lively and Finn groups reported SBF-based polymers through the same synthetic procedures for making SBF monomers, and also highlighted the scalability and translatability of these polymers for membrane separation (Thompson, K. A. *et al. Science* **2020**, 369, 310-315;). In addition, sPIM-SBF is the prototypical PIM polymer with exceptional performance as RFB membranes. The structure-property-performance correlations established in the work will provide design rules for further development of sulfonated PIMs, for example, through alternative synthetic chemistries towards more cost-effective PIMs. *We have added the following explanation in the revised manuscript:*

Revised text “(Manuscript, page 15-16) **Low-cost membrane separators would have significant practical benefits for the widespread deployment of large-scale energy storage technologies. Although, our small-scale, multi-step synthesis of sPIM-SBF would not be cost-effective as it stands, the synthesis is scalable (e.g., there is no requirement for chromatographic separation of intermediates) and the cost per unit mass of the polymer would benefit greatly from large-scale production. Based on a simplified cost analysis (Supplementary Table 13), sPIM-SBF membranes are promising as a more economically competitive alternative to Nafion membranes. In addition, we anticipate that the sPIM-SBF would be used as the thin selective layer (< 1µm) on a cheap microporous support, therefore, the important cost per unit area of membrane will be reasonable. Our sulfonation method will also be applicable to other PIMs which will allow for optimization of materials for more economically competitive membranes. ...**”

Revised note in supplementary information:

The production cost of sPIM-SBF-1.40 was estimated as an example based on the cost of raw materials, reagents and catalysts used in the synthesis (cost data were accessed on websites of common chemical vendors on 01 Dec. 2021). The final materials cost would be £ 2.4 per gram for sPIM-SBF polymers and £ 120 per m² for sPIM-SBF membranes (50 µm thick). It should be noted that this simple estimation does not include the costs associated with manufacturing and operational costs. We envision that large-scale synthesis and manufacturing will further reduce the cost of these polymer membranes.

Supplementary Table 13 | Cost analysis of sPIM-SBF polymer production. The yield of intermediates/product are reported in brackets.

Raw Materials	Price in £ & vendors	Intermediates
Veratrole (1 equiv., 50.0 g)	£ 0.05/g Alfa Aesar	 Compound 1 (75 %)
Bromine (1 equiv., 57.8 g)	£ 0.08/g Alfa Aesar	
Benzene boronic acid (1 equiv., 44.1 g)	£ 0.19/g Acros Organics	
Pd(PPh ₃) ₄ (0.3 % mol, 1.2 g)	£ 7.4/g Fluorochem	 Compound 2 (90 %)
Compound 1 (1 equiv., 58.1 g)		
Bromine (1 equiv., 43.3 g)	£ 0.08/g Alfa Aesar	
Compound 2 (1 equiv., 40.5 g/71.2 g in total)		

n -butyl lithium 2.5 M in hexane (1.05 equiv., 58.2 mL)	£ 0.08/mL Acros Organics	 Compound 3 (76 %)
Methanesulfonic acid (22 w/v%, 180 mL)	£ 0.09/mL Acros Organics	
Compound 2 (1 equiv., 30.7 g)		 Compound 4 (70 %)
Compound 3 (1 equiv., 25.3 g)		
n -butyl lithium 2.5 M in hexane (1.05 equiv., 44.2 mL)	£ 0.08/mL Acros Organics	
Boron tribromide (3 equiv., 55.4 g)	£ 0.40/g Merck Life Sciences	 PIM-SBF (90 %)
Compound 4 (1 equiv., 28.0 g)		
2,3,5,6-tetrafluoroterephthalonitrile (1 equiv., 14.7 g)	£ 0.14/g Fluorochem	 sPIM-SBF-1.40 (98 %)
PIM-SBF (1 equiv., 33.2 g)		
Trimethylsilyl chlorosulfonate (1.7 equiv., 17.4 mL)	£ 1.1/mL Merck Life Sciences	
Total cost: £ 95.6	Yield: 40.2 g	Price: £ 2.4/g

2) One of the advantages of the Nafion membrane is the easiness of its full recover for subsequent uses. The indication that the PIM membranes afford stable battery operations over 2000 charge and discharge cycles does not guarantee that the membranes are recoverable. This should be discussed.

Response: We thank the reviewer for the comment. We performed additional experiments to confirm the recyclability of used sPIM-SBF membranes. The examined samples are post-cycling membranes that were then stored in supporting electrolyte solutions (pH = 9) after being disassembled from batteries before lockdown due to COVID-19 pandemic in March 2020 (cycling performance shown in Fig. 4). No sign of degradation or structural change can be observed from NMR, FT-IR spectra and GPC traces (supplementary Fig. 38). These membranes remain mechanically robust after being bent or folded repeatedly. Moreover, they are readily soluble in DMSO and can be re-processed into robust free-standing membranes (supplementary Fig. 39). Further, we have confirmed the stable ionic conductivity of sPIM-SBF membranes through ageing tests of ~400 days (supplementary Fig. 22). These indicate that used PIM membranes can either be directly re-used in a new battery cell or be re-dissolved to cast a fresh film. *We have added the following explanation in the revised manuscript:*

Revised text“(Manuscript, page 16)...Furthermore, the ease of recovery of membranes from end-of-life batteries for subsequent uses constitutes a critical part in the life cycle assessment of a new membrane product. We characterized, by NMR, FT-IR and GPC, sPIM-SBF membranes that were initially used in low-concentration RFBs for more than 2000 cycles (performance shown in Fig. 4) and then stored in supporting electrolyte solutions for around 20 months. No sign of degradation or structural change can be observed from these measurements (Supplementary Fig. 38). In addition, these membranes remain mechanically robust after being bent or folded repeatedly. Moreover, used sPIM-SBF membranes remain readily soluble in DMSO and can then be re-processed into robust free-standing membranes (Supplementary Fig. 39). These results indicate that PIM membranes from spent RFB cells could either be directly re-used in a new battery cell or be re-dissolved to cast a fresh film.

Revised figure in supplementary information:

Supplementary Fig. 38 | Chemical stability of sPIM-SBF membranes after RFB full cell cycling tests. a&d, liquid ^{13}C NMR, b&e, FT-IR and c&f, GPC of sPIM-SBF-1.40 membrane sPIM-SBF-1.67 membrane after being disassembled from RFB full cell cycling tests for around 46 months. 1) is the original polymer, 2) is the edge of used membranes without contacting with RFB electrolytes and 3) is the effective area of used membranes working in RFB cells, respectively. No structural change is observed in NMR and FT-IR spectra, and no degradation of polymer chains is observed according to GPC traces, indicating the good chemical stability of sPIM-SBF after long-term RFB cycling tests.

Supplementary Fig. 39 | Mechanical stability and solution-processability of sPIM-SBF membranes after RFB full cell cycling tests. a-d, Photographs of sPIM-SBF-1.40 and sPIM-SBF-1.67 membrane after being disassembled from RFB full cell cycling tests for around 20 months. The post-cycling sPIM-SBF membranes maintain their integrity after being bent or fold repeatedly, showing excellent mechanical robustness. e, sPIM-SBF-1.40 and sPIM-SBF-1.67 solution by redissolving used membranes in DMSO. f, sPIM-SBF-1.40 and sPIM-SBF-1.67 membranes fabricated using recycled membrane solutions. The repeatable solution-processability of used sPIM-SBF membranes after battery tests confirm their recyclability.

3) The sulfonated polymers are well characterized, no doubt. However, characterization of the 2,6-DPPAQ along the cycles (beginning, middle and final) would be helpful to the interpretation of the viability of the RFB working with the PIM-SBF membrane. This is partially done with data in Supplementary Fig. 26, but not under working conditions. It is also specifically mentioned on page 12: “The analysis of the positive and negative electrolyte solutions after battery cycling, using NMR spectroscopy, cyclic voltammetry, and ICP showed that the decomposition of these molecules in near neutral pH conditions is relatively slow.”, but then it reports to fig 4 e and f which are schematically and to Supplementary Fig. 36 that shows the CV profiles of redox active molecules after RFB full cell cycling tests. These seem to show some level of changes but are not directly compared to the initial CVs.

Response: We agree with the reviewer that the stability of redox-active species is of crucial importance for ensuring long lifetime of flow battery systems, and that characterization of redox-active species along the cycles would provide useful information. 2,6-DPPAQ was initially reported by Aziz and Gordon groups (Ji, Y. *et al. Advanced Energy Materials* **2019**, 9, 1900039). Critical properties of 2,6-DPPAQ had been thoroughly studied in their work, particularly the chemical and electrochemical stability. Based on a combination of characterization techniques (e.g., NMR, CV, UV-Vis, symmetric and full cell testing), 2,6-DPPAQ was found to be the most stable redox-active organic molecules for aqueous flow battery in their work. We employed the same testing conditions as the Aziz and Gordon work, including electrolyte pH (*i.e.*, 9) and cut-off voltages (0.5 - 1.5 V). Hence, the stability of 2,6-DPPAQ under working conditions would be comparable to that reported previously. Nevertheless, we performed CV measurement for the fresh positive and negative electrolyte solutions (*i.e.*, initial CV) and made comparisons with post-cycling electrolyte solutions. A new peak was found in 2,6-DPPAQ electrolyte solutions after battery cycling (Supplementary Fig. 37 a&b) that corresponds to crossover $K_4Fe(CN)_6$. No other change can be found in the CV measurements.

Revised figure in supplementary information:

Supplementary Fig. 37 | CV profiles of redox active molecules before and after RFB full cell cycling tests. a, CV profiles of 2,6-DPPAQ anolyte before and after battery cycling. **b,** Enlarged views from **a** to show permeated $K_4Fe(CN)_6$ redox peaks. **c,** CV profiles of $K_4Fe(CN)_6$ catholyte before and after battery cycling. **d,** Enlarged views from **c** showing no obvious 2,6-DPPAQ permeation.

4) Being the work essentially focused in a new membrane, it would be advisable to make the comparison with the Nafion membrane with a vanadium RFB, in addition to the 2,6-DPPAQ, $K_4Fe(CN)_6$ pair.

Response: Based on the diverse chemistry of flow battery systems, membrane separators should be designed and tailored to meet the specific requirements of each individual RFB chemistry, including acidic vanadium flow battery, neutral or alkaline aqueous organic RFBs, semi-solid RFBs and metal-air RFBs (Park, M. *et al. Nature Reviews Materials* **2016**, 2, 1 - 18). It is challenging to find a one-size-fits-all solution and develop one specific type of membrane separators that finds universal applications in all types of flow battery systems.

Despite vanadium flow batteries being the most mature flow battery technology, their market penetration is limited by the high cost of Nafion membranes and vanadium. Very few hydrocarbon-based membranes remain stable in vanadium flow batteries due to the presence of highly oxidative VO_2^+ . In contrast, these low-cost membranes can maintain their long-term chemical stability and low ionic resistance in aqueous organic flow batteries, where Nafion membranes are not selective enough to avoid crossover-induced capacity fade. We envision that sPIM-SBF membranes are likely not sufficiently stable for long-term operation of vanadium flow batteries (e.g., cleavage of the dioxin-linkages and hydrolysis of nitrile groups), but we have demonstrated their superior ionic conductivity, unprecedented selectivity towards organics, as well as stability and recyclability in near-neutral pH flow battery, which would advance the development and commercialization of aqueous organic flow batteries. We have added the following discussion to the main text.

Revised text “(Manuscript, page 16)... In recent years, a wide range of flow battery chemistries have been developed beyond conventional vanadium flow battery, such as neutral or alkaline aqueous organic RFBs, semi-solid RFBs, metal-air RFBs, and non-aqueous RFBs²⁻⁷. It is challenging to find a one-size-fits-all solution and develop one specific type of membrane separators that finds universal

applications in all types of flow battery systems. Membrane separators should be designed and tailored to meet specific requirements of each individual RFB chemistry. We envision that sPIM-SBF membranes are likely to be not sufficiently stable for long-term operation of vanadium flow batteries (e.g., cleavage of the dioxin-linkages and hydrolysis of nitrile groups), but we have demonstrated their superior ionic conductivity, unprecedented selectivity towards organics, as well as stability and recyclability in near-neutral pH flow battery, which would advance the development and commercialization of aqueous organic flow batteries.

5) Indication that pH 9 is near neutral pH is erroneous and should be avoided. Neutral pH is 7 -7.5; 9 is an alkaline pH value.

Response: We understand the reviewer's comment that the pH value of 9 is generally defined as an alkaline value based on Brønsted–Lowry acid–base theory (Petrucci, R.H., Harwood, W.S. & Herring, F.G. *General Chemistry* Prentice-Hall **2002**, 8th edition, 666). In aqueous RFB settings, however, the pH value of 9 can be regarded as a near-neutral pH value compared with strongly alkaline solutions with pH over 14. Actually, it is commonly found in RFB publications that pH 9 or even pH 12 is classified as the near-neutral region (Ji, Y. *et al. Adv. Energy Mater.* **2019**, 9, 1900039; Jin, S. *et al. Adv. Energy Mater.* **2020**, 2000100; Kwabi, D. G *et al, Joule* **2**, **2018**, 1894 - 1906).

Classic aqueous organic flow batteries are operated at a pH=14 (e.g., 2,6- dihydroanthraquinones), while most organic species undergo a number of side reactions that lead to irreversible loss of battery capacity at such a high pH condition. Hence, as an important research direction, efforts have been made to reduce the pH of the electrolyte solution towards neutral or near-neutral pH conditions. As reported by Aziz and Gordon groups, 2,6-DPPAQ suffers from severe alkyl chain cleavage side reactions at pH 14 while becoming more stable at pH 12, and exhibits adequate chemical stability at near neutral pH of 9 (Ji, Y. *et al, Adv. Energy Mater.* **2019**, 9, 1900039). Therefore, the operating pH of 9 is classified as near-neutral pH, where redox-active molecules show greatly improved stability as compared with that in strong alkaline condition (e.g., pH 14).

6) BET (Brunauer–Emmett–Teller) surface should not always be abbreviated

Response: Full words of BET has been added when it was first appeared in the main text.

Revised text “(Manuscript, page 6)...although the apparent Brunauer–Emmett–Teller (BET) surface area decreases from 692 m² g⁻¹ to 482 m² g⁻¹ on increasing the degree of sulfonation (Fig. 2d)...”

7) In the Supplementary information section when the battery lifetime is predicted an energy storage capacity value is given of 2kWh for an active surface area of 1m². How the value of energy storage capacity is obtained should be better elucidated.

Response: The energy storage capacity of 2 kWh is one of our assumptions/targets in the lifetime estimation calculations, instead of any experimental value. Based on the materials utilization ratio and capacity decay rate as measured in our laboratory-scale cells, we calculated how much electrolyte would be needed to achieve an energy storage capacity of 2 kWh as well as how long the battery stack would remain in service before reaching the loss of 20% of the total capacity.

Reviewer #2 (Remarks to the Author):

The authors report on a nano-porous ion-selective membrane for redox flow batteries using large organic molecules as the active species, operating with different electrolytes on either side of the membrane. They claim low crossover of active species through size exclusion as well as Donnan exclusion. They report characterization data of the membrane as well as data obtained by cycling a flow battery. The polymer chemistry reported as well as the polymer characterization is sound. However, there are several aspects of this paper that detract from its publication in Nature Communications:

1. The reported conductivities are quite low in spite of measurement in KCl solution.

Response: We respectfully disagree with the reviewer's comment. The values reported in this work correspond to **potassium ionic conductivity** not proton conductivity, which is normally much higher, due to the higher ionic mobility of proton ($36.23 \times 10^{-8} \text{ m}^2 \text{ s}^{-1} \text{ V}^{-1}$) relative to potassium ($7.62 \times 10^{-8} \text{ m}^2 \text{ s}^{-1} \text{ V}^{-1}$). In fact, conductivity with values in the range of 0.02-0.04 S cm^{-1} for sPIM-SBF membranes represents the highest level for potassium conduction as compared to the state-of-the-art ion exchange membranes and ion sieving membranes (Table 1, Supplementary Table 8 and Table R1).

Table R1. Summary of state-of-the-art RFB membranes and their ionic conductivity

Membranes	Ionic Conductivity (S cm^{-1})	Ferri/ferrocyanide permeability ($\text{cm}^2 \text{ s}^{-1}$)	Thickness (μm)	Author/year
Nafion 212	1.1×10^{-2} ^a	1.8×10^{-8} (9.7×10^{-10}) ^f	50	Tan/2020 ¹
AO-PIM-1	4.8×10^{-3} ^a 2.3×10^{-3} ^b	1.7×10^{-9} (2.2×10^{-10}) ^f	50	
PIM-EA-TB	4.4×10^{-4} ^a 2.9×10^{-4} ^b	1.0×10^{-10} (1.1×10^{-11}) ^f	50	
Nafion 117	8.6×10^{-3} ^b	8.4×10^{-10} (1.7×10^{-9}) ^f	178	Zuo/2020 ²
SPX-BP-0.95	9.7×10^{-3} ^c	6.8×10^{-12} (6.5×10^{-10}) ^f	145	Baran/2019 ³
Nafion 212	7.5×10^{-3} ^c	1.8×10^{-12}	55	
Celgard 3501	8.7×10^{-3} ^c	2.2×10^{-7}	25	
AquaPIM-1	7.8×10^{-3} ^c	2.6×10^{-10}	102	Luo/2019 ⁴
Selemon CSO	3.9×10^{-3} ^b	(4.4×10^{-9})	100	
Fumasep E610	1.0×10^{-3} ^e	4.4×10^{-12}	~20	Kwabi/2018 ⁵
Nafion 212	6.6×10^{-3} ^c	4.5×10^{-9}	50±2	De Porcellinis /2018 ⁶
Porous PES/SPEEK-P0	2.9×10^{-2} ^{b'}	1.5×10^{-8}	65±3	Yuan/2018 ⁷
Porous PES/SPEEK-P15	1.4×10^{-2} ^{b'}	1.6×10^{-9}		
Porous PES/SPEEK-P20	3.6×10^{-2} ^{b'}	9.7×10^{-9}		
Porous PES/SPEEK-P25	4.3×10^{-2} ^{b'}	1.1×10^{-8}		
Graphene oxide	1.5×10^{-3} ^{a'} 6.5×10^{-4} ^{d'}	2.0×10^{-10}	~40	Zhang/2018 ⁸
Nafion 115	1.2×10^{-2} ^d	2.3×10^{-9}	150	This work
sPIM-SBF-0.53	2.7×10^{-3} ^d	5.8×10^{-12}	150	
sPIM-SBF-0.98	1.1×10^{-2} ^d	1.8×10^{-11}	148	
sPIM-SBF-1.40	2.0×10^{-2} ^d	2.0×10^{-10}	156	
sPIM-SBF-1.67	3.1×10^{-2} ^d	2.2×10^{-9}	162	
sPIM-SBF-1.86	3.9×10^{-2} ^d	7.5×10^{-9}	182	

^a Measured in 1 M NaOH or ^{a'} 2 M NaOH aqueous solution.

^b Measured in 1 M NaCl or ^{b'} 0.5 M NaCl aqueous solution.

^c Measured in 1 M KOH aqueous solution.

^d Measured in 1 M KCl or ^{d'} 2M KCl aqueous solution.

^e Measured in 1 M K⁺ aqueous solution.

^f Value in brackets represent permeability data derived from RFBs in operation.

References:

1. P. Zuo, Y. Li, A. Wang, R. Tan, Y. Liu, X. Liang, F. Sheng, G. Tang, L. Ge, L. Wu, Q. Song, N. B. McKeown, Z. Yang and T. Xu, *Angew. Chem. Int. Ed.*, 2020, **59**, 9564-9573.

2. R. Tan, A. Wang, R. Malpass-Evans, R. Williams, E. W. Zhao, T. Liu, C. Ye, X. Zhou, B. P. Darwich, Z. Fan, L. Turcani, E. Jackson, L. Chen, S. Y. Chong, T. Li, K. E. Jelfs, A. I. Cooper, N. P. Brandon, C. P. Grey, N. B. McKeown and Q. Song, *Nature materials*, 2020, **19**, 195-202.
3. M. J. Baran, M. N. Braten, S. Sahu, A. Baskin, S. M. Meckler, L. Li, L. Maserati, M. E. Carrington, Y.-M. Chiang, D. Prendergast and B. A. Helms, *Joule*, 2019, **3**, 2968-2985.
4. J. Luo, B. Hu, C. Debruler, Y. Bi, Y. Zhao, B. Yuan, M. Hu, W. Wu and T. L. Liu, *Joule*, 2019, **3**, 149-163.
5. D. G. Kwabi, K. Lin, Y. Ji, E. F. Kerr, M.-A. Goulet, D. De Porcellinis, D. P. Tabor, D. A. Pollack, A. Aspuru-Guzik, R. G. Gordon and M. J. Aziz, *Joule*, 2018, **2**, 1894-1906.
6. D. De Porcellinis, B. Mecheri, A. D'Epifanio, S. Licocchia, S. Granados-Focil and M. Aziz, *J. Electrochem. Soc.*, 2018, **165**, A1137.
7. Z. Yuan, X. Liu, W. Xu, Y. Duan, H. Zhang and X. Li, *Nature communications*, 2018, **9**, 1-11.
8. L. Zhang, Y. Ding, C. Zhang, Y. Zhou, X. Zhou, Z. Liu and G. Yu, *Chem*, 2018, **4**, 1035-1046.

2. Most data is presented without any error bars (something which absolutely needs to be remedied)

Response: We appreciate the reviewer for the suggestion to include error bars in figures. We have updated relevant figures and provided/highlighted the repeated details of corresponding measurements in method section.

Revised figures in main text:

Fig. 2 | Synthesis and characterisation of the sPIM-SBF membranes. c, IEC and skeletal density of PIMs as a function of molar equivalent of TMSCS and PIM-SBF. Error bars are standard deviations derived from the measurements of three different samples for IEC or from ten measurements of one sample for skeletal density.

Fig. 3 | Ion conduction and molecular sieving in membranes. g, KCl permeation rate and KCl/K₄Fe(CN)₆ selectivity of sPIM-SBF and Nafion 115 membranes.

Revised figures and tables in supplementary information:

Supplementary Fig. 8 | Water uptake of sPIM-SBF polymers. b, Aqueous KCl uptake and linear swelling ratio of sPIM-SBF membranes.

Supplementary Table 4 | Physical properties of PIM-SBF and sPIM-SBF polymers.

	IEC (mmol g ⁻¹)	SO ₃ K per SBF unit (/)	S _A _{BET} ^a (m ² g ⁻¹)	Micropore volume ^b (cm ³ g ⁻¹)	Skeletal density (g cm ⁻³)	Envelop density (g cm ⁻³)	Electrolyte uptake (wt.%)	Linear swelling ratio (%)
PIM-SBF	-	-	736	0.227	1.3269 ± 0.0044	1.06	10.9 ± 0.4	0.6 ± 0.02
sPIM-SBF-0.53	0.53±0.01	0.28±0.01	692	0.217	1.3515 ± 0.0029	0.791	26.9 ± 2.2	6.5 ± 0.3
sPIM-SBF-0.98	0.98±0.09	0.56±0.06	634	0.203	1.3786 ± 0.0025	0.991	44.6 ± 1.8	9.0 ± 0.5
sPIM-SBF-1.40	1.40±0.07	0.84±0.05	538	0.182	1.4261 ± 0.0016	0.994	61.5 ± 2.6	15.6 ± 1.1
sPIM-SBF-1.67	1.67±0.16	1.04±0.12	490	0.146	1.4416 ± 0.0026	1.09	96.8 ± 3.9	24.1 ± 0.9
sPIM-SBF-1.86	1.86±0.02	1.20±0.02	482	0.167	1.4802 ± 0.0022	1.23	101 ± 4.0	32.0 ± 1.8

^aS_A_{BET} was obtained by nitrogen adsorption in the range of P/P₀ = 0.001-0.1.

^bMicropore volume was obtained from nitrogen adsorption based on the NLDFT model.

Supplementary Fig. 25 | Selective ion permeation of common salts through sPIM-SBF and Nafion 115 membranes with thickness of around 150 μm. The feed solution is 0.4 M KCl, 0.4 M KNO₃, 0.2 M K₂SO₄, 0.2 M K₂CO₃ or 0.1 M K₄Fe(CN)₆ aqueous solution with varied hydrated diameters of anions (Cl⁻ = 6.64 Å, NO₃⁻ = 7.2 Å, SO₄²⁻ = 7.8 Å, CO₃²⁻ = 8.4 Å, Fe(CN)₆⁴⁻ = 9.0 Å).

= 6.70 Å, SO₄²⁻ = 7.58 Å, CO₃²⁻ = 7.88 Å, Fe(CN)₆⁴⁻ = 8.44 Å¹⁴), respectively, and the permeate side is deionised H₂O. Dash lines are added as guides to the eye.

Supplementary Table 9 | Permeance rate of inorganic salts through sPIM-SBF and Nafion 115 membranes.

Membranes	KCl ^a permeance rate (mol m ⁻² h ⁻¹) ^b	KNO ₃ ^a permeance rate (mol m ⁻² h ⁻¹) ^b	K ₂ SO ₄ ^a permeance rate (mol m ⁻² h ⁻¹) ^b	K ₂ CO ₃ ^a permeance rate (mol m ⁻² h ⁻¹) ^b
Nafion 115	0.14 ± 0.02	0.2 ± 0.05	0.025 ± 0.007	0.017 ± 0.003
sPIM-SBF-0.53	0.054 ± 0.016	0.04 ± 0.007	0.000097	0.000053
sPIM-SBF-0.98	0.12 ± 0.04	0.12 ± 0.02	0.0007	0.00039
sPIM-SBF-1.40	0.4 ± 0.08	0.33 ± 0.09	0.0055 ± 0.0009	0.0021 ± 0.0005
sPIM-SBF-1.67	0.93 ± 0.16	0.94 ± 0.14	0.019 ± 0.003	0.011 ± 0.002
sPIM-SBF-1.86	1.8 ± 0.15	1.8 ± 0.31	0.059 ± 0.010	0.038 ± 0.004

^a0.4 M KCl, 0.4 M KNO₃, 0.2 M K₂SO₄, or 0.2 M K₂CO₃ as a feed solution and deionised H₂O as a permeate solution.

^bPermeation rate through sPIM-SBF and Nafion membranes are calculated by Fick's first law.

Revised text:

“(Manuscript, page 23)...The mean values and standard deviation of skeletal densities were determined on a series of 10 measurements...”

“(Manuscript, page 24)... Average values and standard deviation of the IEC were determined on 3 specimens...”

“(Manuscript, page 24)... The mean values and error bars are derived from three measurements based on three different membranes.”

“(Manuscript, page 25)... Error bars are standard deviations derived from 3 SR measurements based on 3 different samples.”

“(Manuscript, page 25)... The mean values and error bars of ion permeation rate were derived from 3 individual measurements based on 3 different samples.”

3. The conclusion of Donnan exclusion coupled with size-exclusion does not appear to be supported by evidence. The effect appears to be purely that of size exclusion, which is well known and has been previously reported.

Response: We agree with the reviewer that size-sieving, instead of Donnan exclusion, is the main feature of sPIM-SBF membranes that gives high performance and distinguishes them from previously reported polymers.

The size exclusion mechanism dominates the selective transport of anions for sPIM-SBF membranes. A size-exclusion cut-off of ~7 Å was found for sPIM-SBF membranes among a range of anions (*i.e.*, Cl⁻, NO₃⁻, SO₄²⁻, CO₃²⁻, and Fe(CN)₆⁴⁻), which agrees well with the pore size distribution of sPIM-SBF as measured by gas sorption and molecular simulation. In contrast, the size sieving phenomenon was not obvious for commercial benchmark Nafion® membranes that possess similar amount of sulfonate groups (Supplementary Fig. 24, Fig. 25, Table 9 and Table 10).

We attribute the superior ion sieving properties of sPIM-SBF membranes to the well-defined ion transport pathways as well as their polymer chains of extreme rigidity. It has been previously reported that semi-rigid glassy polymers show improved selectivity (*e.g.*, sPSF, sPEAK, sDAPP, s-tripPEEK) as compared to Nafion. However, these polymers have negligible microporosity and the formation of ion transport channels still rely on the uncontrolled microphase separation between different polymer chain segments. **Importantly, one of our design strategies is the use of PIM polymers to provide rigid backbones and well-defined micropores with fixed chain conformation so as to arise from microporosity with narrow size distribution.** *We have revised text in the revised manuscript (shown below).*

Revised text “(Manuscript, page 11)...Indeed, such low permeability values of ferrocyanide and 2,6-DPPAQ can be considered as crossover-free are mainly attributable to efficient size sieving through the rigid PIM backbones that restrict the thermal motions that result in the opening of a void with sufficient size for these large redox active species to move between free volume elements. Whilst the size-sieving is the primary reason for the highly selective ion transport, Donnan exclusion from the negatively charged sulfonate groups may further reduce the permeability of ferrocyanide and 2,6-DPPAQ anions and enhance selectivity (Supplementary Fig. 18).

4. While the cycling numbers seem to be impressive, they also are more a bit disingenuous - the cycle duration seems to be extremely low - perhaps only minutes - while multiple hours / cycle would be the norm. Unless data can be presented at this scale, not many conclusions about suitability can be drawn.

Response: Thank the reviewer for raising this concern. We agree cycle number itself is indeed not informative enough. Hence, following the reviewer' suggestion, we have added cycle time into the figures.

For certain types of RFB systems, such as non-aqueous RFB, hybrid RFB and metal-ion RFB, multiple hours/cycle is the norm due to the high concentration of redox couples (e.g., multiple mole per litre) and/or very low current density that is limited by the sluggish reaction kinetics and slow ion transport. In contrast, for aqueous organic RFBs, lab-scale demonstration of several thousands of cycles with short cycle duration (< 0.5 hour/cycle) is common practice for evaluating new membranes and screening redox-active species (Lin, K. *et al. Science* **2015**, 349, 1529; Janoschka, T. *et al. Nature* **2015**, 527, 78 - 81; Tan, R. *et al. Nat. Mater.* **2020**, 19, 195-202). In our work, laboratory scale RFB cell was constructed to evaluate different membranes within a relatively short duration period at a high current (700 mA for 7 cm² of membrane). As shown in the main figures, a clear cycling-performance difference can be observed within a relatively short duration period for both low-concentration and high-concentration RFBs, where sPIM-SBF-1.40 membrane greatly outperformed benchmark Nafion® membranes. We believe the results have proven the key hypothesis and conclusions of this work.

Revised figures in main text:

Fig. 4 | Low-concentration redox flow battery performance using sPIM-SBF membranes. d, Cycling stability of K₄Fe(CN)₆(0.1 M)/2,6-DPPAQ (0.1 M) RFBs assembled with sPIM-SBF-1.40 (134 μm), sPIM-SBF-1.67 (177 μm) and Nafion 115 membranes at pH 9 and a current density of 100 mA cm⁻².

Fig. 5 | High-concentration redox flow battery performance using sPIM-SBF membranes. e, Cycling stability of RFBs using $K_4Fe(CN)_6$ (1 M)|2,6-DPPAQ (0.4 M) assembled with sPIM-SBF-1.40 or Nafion 115 membranes at pH 9 and a current density of 100 mA cm^{-2} .

Reviewer #3 (Remarks to the Author):

This manuscript reports a new membrane consisting of spirobifluorene-based polymers with intrinsic microporosity (PIM-SBF) for aqueous organic redox flow batteries (AORFBs). By modulating the sulfonation degree of the polymers, the authors attempted to control their porosity and ion conductivity as well as ion selectivity in the crossover. The new membrane fabricated by one polymer (sPIB-SBF-1.40) showed higher ionic conductivity, KCl permeation rate, and anion selectivity than commercial Nafion 115 membrane, which finally led to higher cycle stability in AORFB using 2,6-DPPAQ and K₄Fe(CN)₆. This work is well written, and the authors systematically investigated the structure-property relationship between sulfonation degree of the polymers, their physicochemical properties, and membrane characteristics. However, a few issues should be addressed before acceptance.

Response: We thank the reviewer for positive comments and clearly pointing out the design strategy of our work.

1. The molecular weights and the PDI values of the polymers used in this study should be reported.

Response: As suggested by the reviewer, GPC measurement has been carried out to analyze the molecular weights and PDI values of original PIM-SBF and sPIM-SBF polymers. GPC results of original PIM-SBF show a $M_n = 66.5$ kDa, $M_w = 170.0$ kDa, $M_z = 419.2$ kDa, and a PDI value = 2.56. After sulfonation functionalization, all sulfonated sPIM-SBF polymers maintain high molecular weights, with M_n in the range of 80.0 – 89.7 kDa, M_w in the range of 178.0 – 216.6 kDa, M_z in the range of 390.0 – 443.9 kDa, and relatively narrow molecular weight distributions with PDI values in the range of 2.25 – 2.41. These GPC results suggest negligible change or degradation of PIM-SBF polymeric backbones during sulfonation reaction. On the other hand, the high molecular weights of sPIM-SBF polymers can be further demonstrated by their successful processing into robust and flexible membranes as observed in Fig. 2h. The stress-strain curves show adequate mechanical robustness of these membranes. We added GPC characterization in Supplementary Fig.5 and Table 1, and measurement detail in method section (shown below).

Revised text “(Manuscript, page 23)...Gel permeation chromatography (GPC). GPC measurement of original PIM-SBF was carried out by a GPC MAX 1000 system equipped with two Viscotek columns (CLM3012 LT 5000L) and a RI detector (VE3580) using CHCl₃ (~ 1 mg mL⁻¹) at a flow rate of 1 mL min⁻¹ as an eluent. GPC traces of sPIM-SBF polymers were obtained by a GPC Agilent 1260 system equipped with two PLgel MIXED-C columns (200-2,000,000 g mol⁻¹, 5 μm) and a RI detector using DMF containing 0.1% w/v lithium bromide (~ 5 mg mL⁻¹) at 60 °C at 1 mL min⁻¹ as an eluent. Calibration was performed using narrow dispersity poly(methyl methacrylate) standards.”

Revised figure in supplementary information:

Supplementary Fig. 5 | GPC traces of a, original PIM-SBF and b, sPIM-SBF polymers. GPC results of original PIM-SBF and sPIM-SBF polymers show high molecular weights and relatively narrow molecular weight distributions. These GPC results suggest no or negligible degradation of

sPIM-SBF polymeric backbones by using TMSCS as the sulfonating agent and performing sulfonation functionalization under mild conditions.

Supplementary Table 1 | Molecular weights and PDI values of sPIM-SBF polymers.

Membranes	Mn (kDa)	Mw (kDa)	Mz (kDa)	Mp (kDa)	PDI
PIM-SBF	66.5	170.0	419.2	112.2	2.56
sPIM-SBF-0.53	80.0	178.0	390.0	112.4	2.25
sPIM-SBF-0.98	82.0	190.2	360.0	123.2	2.32
sPIM-SBF-1.40	88.8	207.5	412.1	116.4	2.33
sPIM-SBF-1.67	80.5	213.2	443.9	123.2	2.37
sPIM-SBF-1.86	89.7	216.6	403.5	132.7	2.41

2. The authors characterized the chemical structure of the sulfonated polymers using solid-state ^{13}C NMR. But, sPIM-SBF polymer membranes were fabricated from DMSO solutions. If the polymers were soluble, why did they characterize the polymers in solid-state? If possible, solution-state NMR characterization should be performed. In my opinion, it can provide a more accurate and quantitative characterization for the sulfonation.

Response: Initially, we did attempt liquid state NMR but only got weak signals presumably due to their high viscosity, which is a common observation for PIMs with rigid chains. Hence, we performed solid state NMR instead to characterize the structure of sPIM-SBF. We have now tried liquid state NMR again but adjusted some key parameters to enhance the signal, *i.e.*, increasing the sample concentration (50 mg of sPIM polymers is dissolved in 0.75 mL DMSO- d_6), increasing the temperature up to 333.0 K and increasing the number of scans up to 4096.

The obtained liquid state ^{13}C NMR spectra show similar conclusion as that from solid-state results in terms of sulfonation site, but are indeed more accurate and quantitative. We have replaced the solid state ^{13}C NMR spectra with the liquid state ^{13}C NMR spectra in Supplementary Fig. 3 and provide the detailed characterization information in method section (shown below). We observed significantly broadened peaks in ^1H NMR spectra for sPIM-SBF, which is a distinct characteristic for PIM polymers. The broadened ^1H NMR peaks is related to stronger spin-spin interactions between ^1H nuclei in PIMs with fixed polymeric chain conformation leading to shorter spin-spin relaxation times (T_2) (McKeown, N. B. *Polymer* **2020**, 202, 122736).

Revised text “(Manuscript, page 22)... ^1H and ^{13}C liquid state NMR spectra of original PIM-SBF, sPIM-SBF polymers and 2,6-DPPAQ redox molecules were recorded in deuterated solvents using a Bruker AVA 500 spectrometer (500 and 125 MHz, respectively). PIM-SBF and 2,6-DPPAQ redox molecules were tested at 298 K with number of scans of 8 and 128 for ^1H and ^{13}C NMR. Specifically, high-concentration solution samples of sPIM-SBF polymers (50 mg) in DMSO- d_6 (0.75 mL) were prepared and tested at 333 K. ^1H and ^{13}C spectra were typically compiled from 32 and 4096 scans, respectively.”

Revised figure in supplementary information:

Supplementary Fig. 3 | Liquid state ^{13}C NMR spectra and peak assignment of sPIM-SBF polymers. **a**, Chemical structure of sPIM-SBF polymers. **b**, Liquid state ^{13}C NMR spectra of PIM-SBF and sPIM-SBF polymers with varied degree of sulfonation. The resonance at 144.8 ppm ($4'$) was attributed to carbon environment directly bound to sulfonate groups. Gradually increased resonance at 148.2 ppm ($4'$) and decreased resonance at 128.0 ppm (4) indicated that the degree of sulfonation could be controlled by regulating the molar ratio between TMSCS and PIM-SBF in the reaction⁴.

Fig. S1 | Liquid state ^1H NMR spectra of sPIM-SBF polymers.

3. The raw data for all NMR measurements should be provided in the supplementary information.

Response: We thank the reviewer for pointing out the raw data omission of NMR measurements. The ^1H NMR chemical shifts of monomer 2,2',3,3'-tetramethoxy-9,9'-spirobifluorene (SBF-OMe) and the corresponding sulfonated model compound (sSBF-OMe), and the ^{13}C NMR chemical shifts of sPIM-SBF polymers were added in the method section (shown below). To make the display of NMR raw data clearer, all the ^1H and/or ^{13}C chemical shifts of materials were positioned after synthesis procedure in the method section of main text, and their corresponding ^1H and/or ^{13}C spectra were shown in the supplementary information (Supplementary Fig. 2, Fig. 3, and Fig. 28). The NMR characterization of the synthesis from 2,2',3,3'-tetrahydroxy-9,9'-spirobifluorene (SBF) monomer precursors to PIM-SBF polymer has been reported on our published papers (Bezzu, C. G. *et al. Adv. Mater.* **2012**, 24, 5930-5933; Bezzu, C. G. *et al. J. Mater. Chem. A.* **2018**, 6, 10507-10514).

Revised text "(Manuscript, page 22)...**Synthesis of sulfonated model compound (sSBF-OMe).** A solution of TMSCS (230 mM) in CDCl_3 (100 μL) was added to a solution of 2,2',3,3'-tetramethoxy-9,9'-spirobifluorene (SBF-OMe, 46 mM) in CDCl_3 (400 μL) at 273 K. The sample was then *in situ* monitored by ^1H NMR over a period of 12 h at 278 K. **SBF-OMe:** ^1H NMR (600 MHz, CDCl_3): δ (ppm) 7.71 (m, 2H, ArH), 7.34 (s, 2H, ArH), 7.32 (m, 2H, ArH), 7.03 (m, 2H, ArH), 6.66 (m, 2H, ArH), 6.23 (s, 2H, ArH), 4.03 (s, 6H, OCH_3), 3.64 (s, 6H, OCH_3); **sSBF-OMe:** ^1H NMR (600 MHz, CDCl_3): δ (ppm) 7.89 (br m, 4H, ArH), 7.40 (br s, 2H, ArH), 7.10 (br m, 2H, ArH), 6.21 (br s, 2H, ArH), 4.03 (br s, 6H, OCH_3), 3.65 (br s, 6H, OCH_3).

(Manuscript, page 22)...Since sPIM-SBF polymers with varied degree of sulfonation show same characteristic peaks in FT-IR and ^{13}C NMR spectra, the FT-IR and ^{13}C NMR characteristic peaks of sPIM-SBF-1.40 were listed below as an example. ν_{max} (cm^{-1}): 3400, 3060, 2239, 1601, 1435, 1350, 1269, 1180, 1103, 1030, 1011, 868, 740, 663, 623; ^{13}C NMR (126 MHz, CDCl_3): δ (ppm) 148.3, 146.9, 138.6, 128.0, 125.7, 120.1, 111.5, 109.1, 93.0, 64.5."

4. In this study, it was found that the sulfonation clearly decreased the BET surface area and pore volume of the polymers. However, as the sulfonation degree increased, the ion permeation rate also increased (Fig 3). It is surprising that the increasing rate is steeper in the case of larger anions (Table S8). Furthermore, the ion selectivity was lowered in the more sulfonated polymers (Fig 3 and Table S9), despite lower BET surface, pore-volume, and highly negatively charged backbone structures. The authors should provide a more detailed and reasonable explanation for these behaviours.

Response: We must emphasise that the BET surface area measurements were obtained with dried membranes, while the ion permeation rates were measured with water-swollen membranes. With the degree of sulfonation increases from sPIM-SBF-0.53 to sPIM-SBF-1.86, the BET surface area decreases from 692 to 482 $\text{m}^2 \text{g}^{-1}$, and micropore volume from 0.217 to 0.167 $\text{cm}^3 \text{g}^{-1}$. The decrease of BET surface area and narrower pore size distribution is likely due to the bulkiness of sulfonate and stronger inter-/intra-chain interactions. However, presence of hydrophilic micropores in aqueous solution leads to swelling of membranes and the expansion of pore dimension. Despite lower BET surface area and pore volume are found in heavily sulfonated PIMs, these PIMs tend to swell more significantly, as supported by the electrolyte uptake experiments. Hence, the size of percolated micropores in sPIM-SBF of high IEC becomes larger and less selective towards different ions of varied size. *Therefore, we have added the following explanation in the revised manuscript:*

Revised text: "(Manuscript, page 7)...Combining the results from the experimental measurements and molecular simulation, we attribute the decrease of BET surface area and narrower pore size distribution to the bulkiness of the sulfonate groups and stronger inter-/intra-chain interactions. Despite their lower BET surface areas and pore volumes, the sPIM-SBF polymers tend to swell following absorption of electrolyte solution. A greater degree of sulfonation enhances the degree of swelling of membranes and the expansion of the pore dimensions. Hence, moderately sulfonated sPIM-SBF is optimal to maintain the sub-nanometer size of electrolyte-filled micropores to provide the exquisite size-selectivity that favours charge carrier ions over larger redox-active species.

Reviewers' Comments:

Reviewer #1:

Remarks to the Author:

The authors have correctly responded to almost all the questions previously raised. The only questions that are only partially clarified are: "5) Indication that pH 9 is near neutral pH is erroneous and should be avoided. Neutral pH is 7 -7.5; 9 is an alkaline pH value." the response relies on the fact this concept is related to the B-L acid-base concept and that in the field of RFB alkaline pH is close to 14; despite the fact that the near-neutral is indicated, the readers of Nature may consider this has an odd statement. The near-neutral could be maintained but a clear sentence related to the explanation given "Actually, it is commonly found in RFB publications that pH 9 or even pH 12 is classified as the near-neutral region" should be included.

Question 7 "In the Supplementary information section when the battery lifetime is predicted an energy storage capacity value is given of 2kWh for an active surface area of 1m^2 . How the value of energy storage capacity is obtained should be better elucidated." is not totally convincing: "The energy storage capacity of 2 kWh is one of our assumptions/targets in the lifetime estimation calculations, instead of any experimental value. Based on the materials utilization ratio and capacity decay rate as measured in our laboratory-scale cells, we calculated how much electrolyte would be needed to achieve an energy storage capacity of 2 kWh as well as how long the battery stack would remain in service before reaching the loss of 20% of the total capacity."

But the article is of quality and worth of publication in Nat. Comm.

Reviewer #2:

Remarks to the Author:

I have reviewed the responses to the reviewers' comments and have the following additional comments:

1. The authors have made a sincere attempt to address the reviewers' comments. I appreciate this.
2. Regarding reviewer 2 comment on conductivity - the issue was not K ion vs. proton. The issue was more to do with the measurement being performed in KOH (as opposed to in water with the K ion/counter-ion form of the polymer). Under these circumstances, the value seemed quite low. An easy way to clear this up would be to report the conductivity measured with a blank cell (no membrane, in KOH) alongside the measurement with the membrane in the cell.
3. The response to the issue of Donnan exclusion is only partly satisfactory - the data is conclusive in that it is size exclusion. The authors acknowledge this in their response but continue to mention in the paper that Donnan exclusion may contribute. I do not think there is any contribution from Donnan exclusion whatsoever in this system. Hence, I request the authors to amend the manuscript to remove references to Donnan exclusion as a possible mechanism as I feel this would be scientifically inaccurate.
4. The authors have acknowledged that the cycle time (and not just number of cycles) should be reported, and merely stating 1000s of cycles is disingenuous. I am glad to see that total duration is also now mentioned. However, I caution the authors that the reader usually picks up on cycles and hence, it would not be appropriate to in any way, in the manuscript, claim longevity. I agree that multiple cycles with short duration is a good "accelerated test" and this should be clearly stated. No attempt should be made to sensationalize the "number of cycles without capacity fade" as it looks like each cycle is on the order of a few minutes at most. Do note: regardless of chemistry, the only (narrow) market potential for RFBs will be for long-duration (multi-hour cycles) storage. Any other operational manifestation is not likely to be useful.

I feel that if the above comments are properly addressed, this work can move towards publication.

Reviewer #3:

Remarks to the Author:

The authors well addressed all concerns raised by the reviewers, and the manuscript is now in the final stage for acceptance. But, a few minor issues should still be addressed.

1. I agree that the pH condition where the experiments were conducted in this study can be regarded as near-neutral pH. But, to avoid confusion and thus to help readers from broad fields understand better, the authors are suggested to add exact pH value in the parentheses. For example, near-neutral (pH 9). Those can be added in the abstract part and the main text as well.

2. In the discussion part, the authors compared the performance of their new membrane in an organic RFB with the previously reported membranes. As the authors already concluded in this study, its superior performance was attributed to the exceptionally low crossover of active materials through the new membrane. Although the crossover significantly depends on the size of the active materials, most of the active materials used in the previous studies have smaller sizes, for example, 2,6-DHAQ (see Table 1).

For a fair comparison and showing the versatility of the new membrane, the authors are suggested to conduct an ion permeation test or cycle stability test of RFB using 2,6-DHAQ.

Response to reviewers' comments

We are grateful to all reviewers again for their time and additional valuable comments. Below are the point-to-point response.

Reviewers' comments (Blue); Authors' response: black; Revised text in manuscript and ESI (Red).

Reviewer #1 (Remarks to the Author):

The authors have correctly responded to almost all the questions previously raised. The only questions that are only partially clarified are: "5) Indication that pH 9 is near neutral pH is erroneous and should be avoided. Neutral pH is 7-7.5; 9 is an alkaline pH value." the response relies on the fact this concept is related to the B-L acid-base concept and that in the field of RFB alkaline pH is close to 14; despite the fact that the near-neutral is indicated, the readers of Nature may consider this has an odd statement. The near-neutral could be maintained but a clear sentence related to the explanation given "Actually, it is commonly found in RFB publications that pH 9 or even pH 12 is classified as the near-neutral region" should be included.

Response: We appreciate the reviewer's positive comments and valuable suggestion. To make to the definition of near-neutral condition clearer, we have added the following explanation in the revised manuscript:

(Manuscript, page 1, abstract) Importantly, these cation exchange membranes demonstrate ultrahigh conductivity for aqueous salt electrolytes at near-neutral pH (pH = 9), in which both a redox-active anthraquinone and ferrocyanide show long-term stability.

(Manuscript, page 3) In this work, we report new ion-exchange membranes with subnanometer ion transport pathways derived from PIMs and demonstrate their exceptional performance in aqueous organic RFBs operated at near neutral pH conditions (pH = 9, Fig. 1a).

(Manuscript, page 3) Importantly, the membranes demonstrate high conductivity for aqueous electrolytes at near-neutral pH of 9, in which both redox-active anthraquinone and ferrocyanide show long-term stability.

(Manuscript, page 11) In contrast to conventional aqueous organic RFB chemistries operated with highly alkaline solutions (e.g., pH = 14), electrolyte solutions with lower pH (i.e. pH range 9-12), which are commonly termed near-neutral pH in aqueous RFBs, are proven to be beneficial for minimizing the degradation of both redox-active molecules and polymer membranes.

(Manuscript, page 18) In summary, we have demonstrated a new generation of ion-exchange membranes that provide exceptional performance in aqueous organic RFBs operated at near neutral pH conditions (pH = 9).

Question 7 "In the Supplementary information section when the battery lifetime is predicted an energy storage capacity value is given of 2kWh for an active surface area of 1m². How the value of energy storage capacity is obtained should be better elucidated." is not totally convincing: "The energy storage capacity of 2 kWh is one of our assumptions/targets in the lifetime estimation calculations, instead of any experimental value. Based on the materials utilization ratio and capacity decay rate as measured in our laboratory-scale cells, we calculated how much electrolyte would be needed to achieve an energy storage capacity of 2 kWh as well as how long the battery stack would remain in service before reaching the loss of 20% of the total capacity." But the article is of quality and worth of publication in Nat. Comm.

Response: We have added the more detailed explanation on the calculation of energy storage capacity in the revised supporting information.

(Supplementary information, page 47) As demonstrated in this work (Figs. 4 and 5), the typical power density of organic redox flow batteries is 1 kW/m² (100 mW cm⁻²) at 0.1 A cm⁻². Given a small flow battery cell with an active surface area of 1 m² and rated power of 1 kW, and energy storage for 2 h, the energy storage capacity is 2 kWh. The current is equal to the area multiplied by the current density: 0.1 A cm⁻² × 1 m² = 1000 A; The charge is equal to current multiplied by time: 2 h × 1000 A = 7200 s × 1000 C s⁻¹ = 7.2 × 10⁶ C. Moles of electrons = 7.2 × 10⁶ C / 96,485 C mol⁻¹ = 74.6 moles of electrons.

Reviewer #2 (Remarks to the Author):

I have reviewed the responses to the reviewers' comments and have the following additional comments:

1. The authors have made a sincere attempt to address the reviewers' comments. I appreciate this.
2. Regarding reviewer 2 comment on conductivity - the issue was not K ion vs. proton. The issue was more to do with the measurement being performed in KOH (as opposed to in water with the K ion/counter-ion form of the polymer). Under these circumstances, the value seemed quite low. An easy way to clear this up would be to report the conductivity measured with a blank cell (no membrane, in KOH) alongside the measurement with the membrane in the cell.

Response: We agree with the reviewer's concern about the deviation of real membranes conductivity caused by the blank cell filled with aqueous electrolyte. Indeed, we have taken this factor into the consideration and the resistance of blank cells had been subtracted to evaluate the real ionic conductivity of sPIM-SBF membranes as reported in the manuscript. In detail, after the EIS measurement of each membrane-loaded coin cell (Type 2032), the membrane was removed and the otherwise-identical cell without any membrane was re-assembled to obtain the blank resistance. The resistance of all blank cells was similar with values of around $0.100 \Omega \text{ cm}^2$ based on individual 12 measurements.

For apparent ionic conductivity measurement in 1M aqueous KCl, sPIM-SBF membranes with sufficient IEC values (*i.e.*, except sPIM-SBF-0.53) show low area specific resistance (ASR) ranging from 0.244 to $0.698 \Omega \text{ cm}^2$, which are superior to Nafion 115 membrane and even comparable to those of blank coin cells (Table R1). And these low ASR values leads to ultrahigh ionic conductivity with values in the range of $0.01\text{-}0.04 \text{ S cm}^{-1}$ for sPIM-SBF membranes, representing the highest level for potassium ion conduction as compared to commercial benchmark membranes, recently reported state-of-the-art ion exchange membranes and ion sieving membranes¹⁻⁸. To help readers clearly identify the low resistance and high ionic conductivity of our sPIM-SBF membranes, we have added the membrane resistance values in the revised supplementary information:

Table R1 | Area specific resistance (ASR) of blank and membrane-loaded coin cells, and the derived real ionic conductivity (σ) of sPIM-SBF and Nafion membranes.

Polymer	Area specific resistance ($\Omega \text{ cm}^2$)			Apparent ionic conductivity ($10^{-3} \text{ S cm}^{-1}$) ^c
	Blank cells ^a	Membrane loaded cells ^b	Membranes	
sPIM-SBF-0.53	0.110	2.992	2.882	2.7
sPIM-SBF-0.98	0.098	0.796	0.698	11.3
sPIM-SBF-1.40	0.096	0.525	0.429	19.5
sPIM-SBF-1.67	0.101	0.375	0.274	31.2
sPIM-SBF-1.86	0.101	0.345	0.244	39.4
Nafion 115	0.098	0.726	0.628	11.9

^a Assembled blank coin cells filled with 1 M aqueous KCl solution and measured at 30 °C by EIS.

^b Assembled coin cells loaded with membranes and filled with 1 M aqueous KCl solution and measured at 30 °C by EIS.

^c Apparent ionic conductivity derived from ASR measured in 1 M aqueous KCl solution using the equation $\sigma = L/(AR_m)$.

References:

1. Zuo, P. *et al.* Sulfonated Microporous Polymer Membranes with Fast and Selective Ion Transport for Electrochemical Energy Conversion and Storage. *Angew. Chem. Int. Ed.* **59**, 9564-9573 (2020).
2. Tan, R. *et al.* Hydrophilic microporous membranes for selective ion separation and flow-battery energy storage. *Nat. Mater.* **19**, 195-202 (2020).
3. Baran, M. J. *et al.* Design Rules for Membranes from Polymers of Intrinsic Microporosity for Crossover-free Aqueous Electrochemical Devices. *Joule* **3**, 2968-2985 (2019).
4. Luo, J. *et al.* Unprecedented Capacity and Stability of Ammonium Ferrocyanide Catholyte in pH Neutral Aqueous Redox Flow Batteries. *Joule* **3**, 149-163 (2019).
5. Kwabi, D. G. *et al.* Alkaline Quinone Flow Battery with Long Lifetime at pH 12. *Joule* **2**, 1894-1906 (2018).
6. Porcellinis, D. D. *et al.* Communication—Sulfonated Poly (ether ether ketone) as Cation Exchange Membrane for Alkaline Redox Flow Batteries, *J. Electrochem. Soc.*, 2018, **165**, A1137.
7. Yuan, Z. *et al.* Negatively Charged Nanoporous Membrane for A Dendrite-free Alkaline Zinc-based Flow Battery with Long Cycle Life, *Nature communications*, 2018, **9**, 1-11.
8. Zhang, L. *et al.* Enabling Graphene-Oxide-Based Membranes for Large-Scale Energy Storage by Controlling Hydrophilic Microstructures. *Chem* **4**, 1035-1046 (2018).

Revised tables in supplementary information:

Supplementary Table 8 | Membrane resistance (R_m , Ω cm²), ionic conductivity (σ , 10^{-3} S cm⁻¹) and activation energy (eV) of sPIM-SBF and Nafion membranes.

Polymer	R_m in KCl ^a	Apparent σ ^b	R_m in DI water ^c	Intrinsic σ ^d	Activation energy ^e
sPIM-SBF-0.53	2.882	2.7	5.699	1.5	0.14
sPIM-SBF-0.98	0.698	11.3	1.256	5.5	0.10
sPIM-SBF-1.40	0.429	19.5	1.029	9.0	0.09
sPIM-SBF-1.67	0.274	31.2	0.593	13.7	0.08
sPIM-SBF-1.86	0.244	39.4	0.500	16.9	0.07
Nafion 115	0.628	11.9	2.348	3.2	0.12

^a Membrane resistance measured in 1 M aqueous KCl solution at 30 °C by EIS.

^b Apparent ionic conductivity derived from membrane resistance measured in 1 M aqueous KCl using equation $\sigma = L/(AR_m)$.

^c Membrane resistance measured in DI water at 30 °C by EIS.

^d Intrinsic ionic conductivity derived from membrane resistance measured in DI water using equation $\sigma = L/(AR_m)$.

^e Derived from apparent ionic conductivity measured in 1 M aqueous KCl solution.

3. The response to the issue of Donnan exclusion is only partly satisfactory - the data is conclusive in that it is size exclusion. The authors acknowledge this in their response but continue to mention in the paper that Donnan exclusion may contribute. I do not think there is any contribution from Donnan exclusion whatsoever in this system. Hence, I request the authors to amend the manuscript to remove references to Donnan exclusion as a possible mechanism as I feel this would be scientifically inaccurate.

Response: As requested by the reviewer, we have made the following changes in the revised manuscript:

(**Manuscript, page 1, abstract**) The spirobifluorene unit allows exquisite control over the degree of sulfonation of the PIM in order to optimize the rapid transport of small cations, whilst the ion exchanged subnanometer pores effectively prohibit the crossover of large ~~anions and anionic~~ organic molecules via molecular sieving ~~and Donnan exclusion~~.

(**Manuscript, page 10**) Despite the electroneutrality requirement that must be met in a concentration-driven diffusion leading to coupled transport of cations and anions, the transport of anions is controlled by multiple factors, particularly ion size, ion mobility and size sieving effect ~~and Donnan exclusion~~.

(**Manuscript, page 11**) Indeed, such low permeability values of ferrocyanide and 2,6-DPPAQ can be considered as crossover-free and are mainly attributable to efficient size sieving through the rigid PIM backbones which restrict the thermal motions that result in the opening of a void with sufficient size for these large redox active species to move between free volume elements. ~~Whilst the size sieving is the primary reason for the highly selective ion transport, Donnan exclusion from the negatively charged sulfonate groups may further reduce the permeability of ferrocyanide and 2,6-DPPAQ anions and enhance selectivity (Supplementary Fig. 18).~~

4. The authors have acknowledged that the cycle time (and not just number of cycles (should be reported, and merely stating 1000s of cycles is disingenuous. I am glad to see that total duration is also now mentioned. However, I caution the authors that the reader usually picks up on cycles and hence, it would not be appropriate to in any way, in the manuscript, claim longevity. I agree that multiple cycles with short duration is a good "accelerated test" and this should be clearly stated. No attempt should be made to sensationalize the "number of cycles without capacity fade" as it looks like each cycle is on the order of a few minutes at most. Do note: regardless of chemistry, the only (narrow) market potential for RFBs will be for long-duration (multi-hour cycles) storage. Any other operational manifestation is not likely to be useful.

Response: We agree with the reviewer that capacity decay as a function of time should be primarily used for evaluating the performance and stability of flow batteries. Hence, we have made some changes to the main paper and revised the claims of longevity based on cycle numbers.

(Manuscript, page 1, abstract) The new PIM membranes significantly boost battery energy efficiency and peak power density and enable stable operations of concentration-optimized RFBs **for about 120 h in laboratory scale flow cells ~~over 2000 charge and discharge cycles~~**.

(Manuscript, page 3) By pairing the new PIM membranes with neutral-pH electrolytes, our membranes enable efficient and highly stable battery operations **for about 120 h in laboratory scale flow ~~cells over 2000 charge and discharge cycles~~**, demonstrating a significantly improved lifetime of organic flow batteries.

(Manuscript, page 11) Laboratory-scale asymmetric redox flow cells were constructed based on $\text{K}_4\text{Fe}(\text{CN})_6$ (0.1 M) and 2,6-DPPAQ (0.1 M) operating at a pH of 9 in an argon-filled glovebox **for accelerated assessment of membrane performance**.

(Manuscript, page 12) An RFB based on sPIM-SBF-1.40 membrane maintains high energy efficiency and **a very low capacity decay rate of 0.0335% per day (0.0000795% per cycle) for about 120 h** (2100 charge-discharge cycles), which is orders of magnitude lower than that of an otherwise identical RFB using Nafion 115 membrane (2.17% per day and 0.00472% per cycle).

We also added one sentence in the supporting information:

(Supplementary information, page 48) **It should be noted that the above estimation did not consider the degradation of redox species, degradation of membranes, fouling of membranes, pressure of electrolyte solutions, and water migration.**

Reviewer #3 (Remarks to the Author):

The authors well addressed all concerns raised by the reviewers, and the manuscript is now in the final stage for acceptance. But, a few minor issues should still be addressed.

Response: We thank the reviewer for the positive comments and the valuable suggestions.

1. I agree that the pH condition where the experiments were conducted in this study can be regarded as near-neutral pH. But, to avoid confusion and thus to help readers from broad fields understand better, the authors are suggested to add exact pH value in the parentheses. For example, near-neutral (pH 9). Those can be added in the abstract part and the main text as well.

Response: We have added the explanation in the revised manuscript based on the comments from both reviewer #1 and reviewer #3.

(**Manuscript, page 1, abstract**) Importantly, these cation exchange membranes demonstrate ultrahigh conductivity for aqueous salt electrolytes at near-neutral pH (pH = 9), in which both a redox-active anthraquinone and ferrocyanide show long-term stability.

(**Manuscript, page 3**) In this work, we report new ion-exchange membranes with subnanometer ion transport pathways derived from PIMs and demonstrate their exceptional performance in aqueous organic RFBs operated at near neutral pH conditions (pH = 9, Fig. 1a).

(**Manuscript, page 3**) Importantly, the membranes demonstrate high conductivity for aqueous electrolytes at near-neutral pH of 9, in which both redox-active anthraquinone and ferrocyanide show long-term stability.

(**Manuscript, page 11**) In contrast to conventional aqueous organic RFB chemistries operated with highly alkaline solutions (e.g., pH = 14), electrolyte solutions with pH = 9 or even pH = 12 that are commonly classified as near-neutral region in aqueous RFBs are proven to be beneficial for minimizing the degradation of both redox-active molecules and polymer membranes.

(**Manuscript, page 18**) In summary, we have demonstrated a new generation of ion-exchange membranes that provide exceptional performance in aqueous organic RFBs operated at near neutral pH conditions (pH = 9).

2. In the discussion part, the authors compared the performance of their new membrane in an organic RFB with the previously reported membranes. As the authors already concluded in this study, its superior performance was attributed to the exceptionally low crossover of active materials through the new membrane. Although the crossover significantly depends on the size of the active materials, most of the active materials used in the previous studies have smaller sizes, for example, 2,6-DHAQ (see Table 1). For a fair comparison and showing the versatility of the new membrane, the authors are suggested to conduct an ion permeation test or cycle stability test of RFB using 2,6-DHAQ.

Response: We thank the reviewer for the suggestion. We have performed 2,6-DHAQ crossover tests and compared the results with previously reported membranes. The permeability of 2,6-DHAQ for sPIM-SBF membranes are in the order of 10^{-10} cm² s⁻¹, values comparable to those of recently reported ion-sieving membranes and Nafion membranes (Table R2). Based on the crossover tests, sPIM-SBF membranes show higher selectivity towards the bulkier 2,6-DPPAQ than 2,6-DHAQ when compared with Nafion membranes, further validating the size-exclusion mechanism that governs selective ionic and molecular transport across sPIM-SBF membranes. We have made the following changes in the revised manuscript:

(**Manuscript, page 11**) Similarly, diffusion rates of bulky organic molecules (i.e., 2,6-di(3-phosphenoxy)anthraquinone, 2,6-DPPAQ) are low with permeability in the order of 10^{-13} ~ 10^{-12} cm² s⁻¹ (Fig. 3f, Supplementary Fig. 27 and Table 11), while the diffusion rates for smaller 2,6-DHAQ are much faster but the values are still comparable with Nafion and ion-sieving membranes.

(**Manuscript, page 22**) In redox molecule crossover tests, K₄Fe(CN)₆ (0.1 M) or 2,6-DPPAQ (0.1 M) in KCl aqueous solution (1 M, 50 mL, pH = 9.0) was used as a feed solution and blank KCl aqueous solution (1 M, 50 mL, pH = 9.0) was used in the permeate side. 2,6-DHAQ (0.1 M) in KOH aqueous solution (1 M, 50 mL) was used as a feed solution and blank KOH aqueous solution (1 M, 50 mL) was used in the permeate side... The

concentration change of 2,6-DPPAQ or 2,6-DHAQ in the permeate solution was quantitatively detected by the calibrated UV-Vis spectrometer (Supplementary Fig. 26).

Table R2. | Permeability of redox-active molecules and ionic conductivity through state-of-the-art RFB membranes.

Membranes	2,6-DHAQ permeability (cm ² s ⁻¹) ^c	Ferri/ferrocyanide permeability (cm ² s ⁻¹)	Ionic Conductivity (S cm ⁻¹)	Author/year
Nafion 212	6.1×10 ^{-10a}	1.8×10 ^{-8a}	1.1×10 ^{-2a}	Tan/2020 ¹
AO-PIM-1	2.1×10 ^{-9a}	1.7×10 ^{-9a}	4.8×10 ^{-3a}	
PIM-EA-TB	4.9×10 ^{-10a}	1.0×10 ^{-10a}	4.4×10 ^{-4a}	
Nafion 117	6.7×10 ^{-10b}	8.4×10 ^{-10b}	8.6×10 ^{-3c}	Zuo/2020 ²
SPX-BP-0.95	3.9×10 ^{-9b}	6.8×10 ^{-12b}	9.7×10 ^{-3c}	
Nafion 115	1.1×10 ^{-10c}	2.2×10 ^{-9d}	1.2×10 ^{-2d}	This work
sPIM-SBF-0.98	1.1×10 ^{-10c}	2.6×10 ^{-11d}	1.1×10 ^{-2d}	
sPIM-SBF-1.40	8.4×10 ^{-10c}	1.7×10 ^{-10d}	2.0×10 ^{-2d}	
sPIM-SBF-1.67	6.0×10 ^{-9c}	2.3×10 ^{-9d}	3.1×10 ^{-2d}	

^a Measured in 1 M NaOH aqueous solution.

^b Measured in 0.1 M KOH aqueous solution.

^c Measured in 1 M KOH aqueous solution.

^d Measured in 1 M KCl aqueous solution.

References:

1. Tan, R. *et al.* Hydrophilic Microporous Membranes for Selective Ion Separation and Flow-battery Energy Storage. *Nat. Mater.* **19**, 195-202 (2020).
2. Zuo, P. *et al.* Sulfonated Microporous Polymer Membranes with Fast and Selective Ion Transport for Electrochemical Energy Conversion and Storage. *Angew. Chem. Int. Ed.* **59**, 9564-9573 (2020).

Revised figures and table in supplementary information:

Supplementary Table 11 | Permeance rate (mol m⁻² h⁻¹) and permeability (cm² s⁻¹) of redox-active molecules through sPIM-SBF and Nafion membranes.

Membranes	K ₄ Fe(CN) ₆ ^a		2,6-DPPAQ ^a		2,6-DHAQ ^b	
	Permeance rate	Permeability	Permeance rate	Permeability	Permeance rate	Permeability
sPIM-SBF-0.53	2.2×10 ⁻⁷	1.1×10 ⁻¹²	-	-	-	-
sPIM-SBF-0.98	6.4×10 ⁻⁶	2.6×10 ⁻¹¹	-	-	7.1×10 ^{-5c}	2.4×10 ⁻¹⁰
sPIM-SBF-1.40	4.3×10 ⁻⁵	1.7×10 ⁻¹⁰	1.7×10 ⁻⁷	6.3×10 ⁻¹³	8.4×10 ^{-4c}	9.1×10 ⁻¹⁰
sPIM-SBF-1.67	5.5×10 ⁻⁴	2.3×10 ⁻⁹	3.0×10 ⁻⁷	1.3×10 ⁻¹²	1.1×10 ^{-3c}	6.0×10 ⁻⁹
sPIM-SBF-1.86	1.6×10 ⁻³	7.5×10 ⁻⁹	-	-	-	-
Nafion 115	5.1×10 ⁻⁴	2.2×10 ⁻⁹	4.6×10 ⁻⁷	2.0×10 ⁻¹²	2.3×10 ^{-5d}	9.9×10 ⁻¹¹

^a 0.1 M K₄Fe(CN)₆ or 0.1 M 2,6-DPPAQ in 1M KCl at pH 9 as a feed solution and 1 M aqueous KCl at pH 9 as a permeate solution.

^b 0.1 M 2,6-DHAQ in 1M aqueous KOH as a feed solution and 1 M aqueous KOH as a permeate solution.

^c The thickness of sPIM-SBF-0.98, sPIM-SBF-1.40, sPIM-SBF-1.67 and Nafion 115 membranes is 119, 39, 190 μm, respectively. Other permeance rate were evaluated from ~ 150-um-thick sPIM-SBF membranes.

Response to reviewers' comments

Reviewer #1 (Remarks to the Author):

In this second review the authors have managed to respond to the questions previously raised. The article can be published in Nat. Comm.

Response: We appreciate the reviewer's positive comments.

Reviewer #2 (Remarks to the Author):

The authors have addressed the comments well. I am Ok with publication provided the authors make the following minor change:

1. Supplementary Figure 18 - Please remove reference of Donnan exclusion to make it consistent with the rest of the manuscript.

Response: We agree with the reviewer's suggestion. We have removed supplementary figure 18 which is related to Donnan exclusion to make the manuscript more consistent.

Reviewer #3 (Remarks to the Author):

The authors well addressed all issues. I recommend publication of this manuscript now.

Response: We thank the reviewer for the positive comments.

Reviewers' Comments:

Reviewer #1:

Remarks to the Author:

In this second review the authors have managed to respond to the questions previously raised. The article can be published in Nat. Comm.

Reviewer #2:

Remarks to the Author:

The authors have addressed the comments well. I am Ok with publication provided the authors make the following minor change:

1. Supplementary Figure 18 - Please remove reference fo Donnan exclusion to make it consistent with the rest of the manuscript.

Reviewer #3:

Remarks to the Author:

The authors well addressed all issues. I recommend publication of this manuscript now.